# Phonology of Adur Niesu in Liangshan, Sichuan

## Hongdi Ding

Department of Chinese Language Studies (CHL), The Education University of Hong Kong, Tai Po, Hong Kong; hding@eduhk.hk

**Abstract:** This study describes the segmental and suprasegmental phonology of Adur Niesu, a Loloish (or Ngwi) language spoken mainly in Liangshan, Sichuan, southwest China. Phonemically, there are 41 consonants, 10 monophthongs and 1 diphthong in Adur Niesu. All Adur syllables are open. Its segmental changes mainly happen to the vowels, featuring high vowel fricativization, vowel lowering, vowel centralization, vowel assimilation and vowel fusion. It is common for Adur Niesu syllables to be reduced in continuous speech, with floating tones left. There are three main types of syllable reduction: complete reduction including the segment and tone, partial reduction with a floating tone left, and partial reduction with the initial consonant left. Adur Niesu employs tones as an important means for lexical contrast, namely, high-level tone 55, mid-level tone 33, and low-falling tone 21. There is also a sandhi tone 44. There are two types of tonal alternation: tone sandhi and tone change. Tone sandhi occurs at both word and phrasal levels, and is conditioned by the phonetic environment, while tone change occurs due to the morphosyntactic environment. Finally, some seeming tonal alternation is the result of a floating tone after syllable reduction.

**Keywords:** Adur Niesu; phonology; consonants; vowels; tones; tone sandhi; tone change

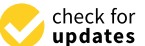



## 1. Introduction

Adur Niesu is a member of the Nisoic (aka. Loloish or Ngwi) subgroup of the Niso–Burmese (i.e., Burmese–Lolo) language group of the Tibeto–Burman languages (Bradley 1997; Lama 2012). It is spoken by about 440,000 people, who are officially recognized as Yi (彝族), residing in mountainous regions in Liangshan (literally 'Cool Mountains'), Sichuan, in southwest China. The Adur Niesu people often call themselves simply Adur, which is said to be the surname of a famous ruling clan living in Butuo (or ndzɿ⁵⁵la³³pu⁴⁴tʰɯ³³) in eastern Liangshan. Adur is often associated with the title ndzɿ³³mo²¹ (lord caste:master) 'highest lord caste' and its variant ndzɿ²¹mo²¹ (lord caste:big) 'big (accomplished) highest lord caste', namely, a³³t̠u³³ndzɿ³³mo²¹ and ndzɿ²¹mo²¹a³³t̠u³³. It is noted that the tone on the morpheme meaning 'lord caste' is different when it is before and after Adur, namely, ndzɿ³³ and ndzɿ²¹. This reflects a tone change that will be discussed in Section 4.4.3. When ndzɿ³³mo²¹ is placed after Adur, without any tone change, it functions as a title, similar to the structure in su³³ga⁵⁵ ma⁵⁵mo²¹ (surname teacher) 'Mr. Suga'. When ndzɿ²¹mo²¹ is placed before Adur, with a tone change, it is a nominal modifier meaning 'big (accomplished) lord caste', similar to dza⁴⁴ndo³³vi³³ su̠³³ga⁵⁵ (food:swallow:type surname) 'Suga, big eater'. The Adur Niesu people mainly live in Butuo (布拖县), Puge (普格县) and Ningnan (宁南县), with some Adur population located along the border with Jinyang (金阳县) and Zhaojue (昭觉县); see Figure 1.

Moreover, Adur people also call themselves Niesu [njɛ³³ su³³]. This autonym is shared by another group of Yi people adjacent to the Adur region, called Suondi Niesu or simply Suondi or Niesu; see Figure 1. Niesu [njɛ³³ su³³] has two meaning-bearing morphemes, namely, [njɛ³³] 'black' and [su³³] 'people', which literally means 'black people'. The population of Suondi Niesu is around 550,000, estimated according to Chen et al. (1985); Gerner (2013) and the 2010 Population Census of Liangshan. Major Suondi-speaking regions are Dechang (德昌县), Huili (会理县), and Puge (普格县) within Liangshan, Miyi (米易县) in

the adjacent city of Panzhihua (攀枝花市) in Sichuan, and Yongren (永仁县) and Yuanmou (元谋县) in Yunnan. Mutual intelligibility between Suondi Niesu and Adur Niesu is relatively high.

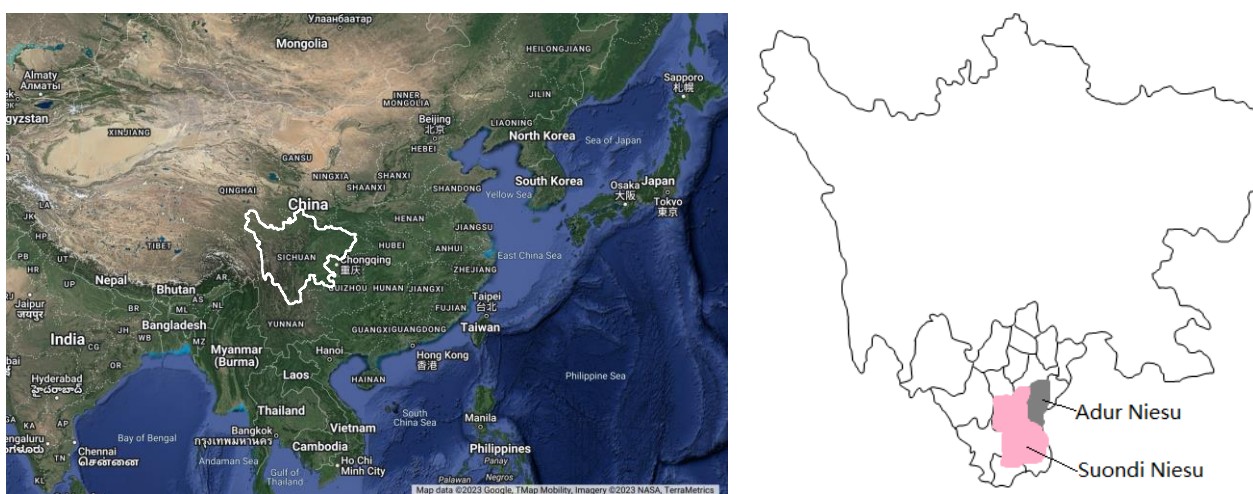

**Figure 1.** Distribution of Adur Niesu.

There are three recent studies about Niesu phonology, mostly focusing on Suondi Niesu in Mahai (2015, 2019) and in Mise (2020). Since Suondi Niesu is very close to Adur Niesu, these works are important references to understand Adur Niesu. But there is still room to improve the accuracy and adequacy of the analysis. Although some phonetic information, i.e., Adur consonants, vowels and tones, are presented in Sun's (2020) construction of an Adur phonetics corpus, there is little research on the phonology. Therefore, this study will contribute to the literature by describing the phonological system of Adur Niesu. The Adur Niesu data presented in this paper are first-hand fieldwork data collected through spontaneous narration and elicitation, mainly based on the Tuojue dialect spoken in central Butuo, Liangshan. The fieldwork in Tuojue (or 拖觉镇), Butuo, started in 2018 and there have been five trips so far; each trip lasted for about two months. The two main consultants are Adur Niesu native speakers who are in their 30s. They started to learn Chinese after they were 10 years old in school and became fluent in Chinese around the age of 18. The data presented in the paper were also cross checked with elder speakers aged from 50 to 70 in Butuo, Liangshan. Although a series of studies have been devoted to the labiovelar sounds in Adur Niesu (i.e., kp, kph, gb, gb, ŋm) (Pan 2001; Matisoff 2006; Hajek 2006; Bradley 2008), such sounds are not found in the Tuojue dialect.[1]

## 2. About Adur Niesu

Based on the subgrouping in Hammarström et al. (2022), Adur Niesu is a verb–final syllable–tone Burmo–Qiangic language; see Figures 2 and 3. Its morphology is largely isolating. A large number of phonemic consonants in Adur Niesu are generated by voicing, aspiration and prenasalization. The grammatical function of Adur Niesu is mainly conveyed by using clitics and postpositions. Property-denoting modifiers follow the head noun. However, noun and genitive modifiers precede the head noun. Tense is not a grammatical category in Adur Niesu. The relation of the event time to some temporal reference point is expressed by lexical means, such as $a^{21}\eta u^{33}$ 'now', $a^{21}\eta i^{55}$ 'the past' and $i^{21}s\epsilon^{21}s\textcommabelow{l}^{44}a^{33}\textbarl o^{44}$ 'the ancient past'. Its aspectual classes are expressed strictly analytically, by verbal enclitics, TAM auxiliaries, and periphrastic constructions. Adur Niesu forms its yes/no questions by reduplicating the last syllable of the verb or auxiliary. It is topic-prominent, frequently employing topic–comment constructions.

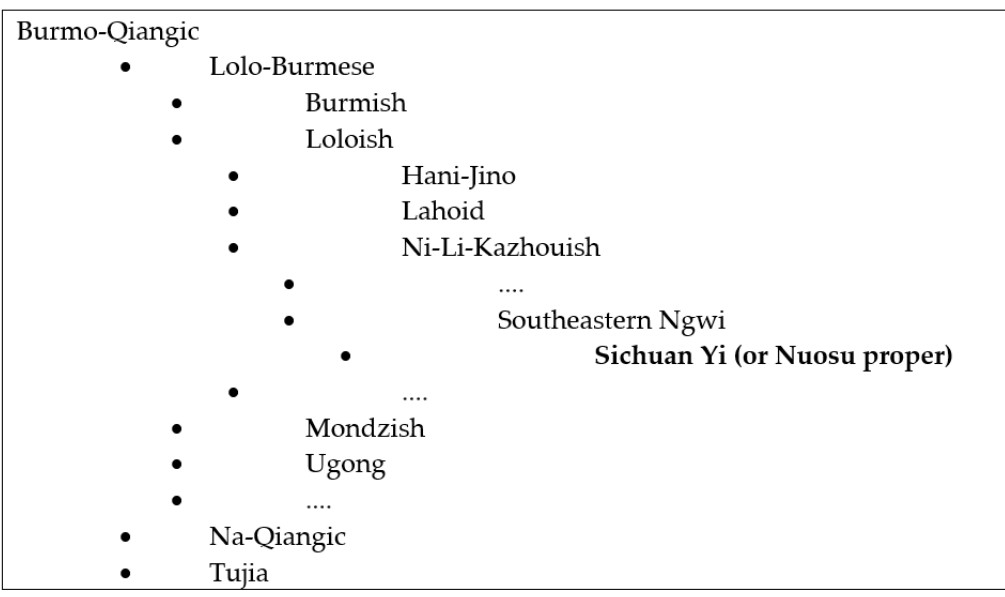

**Figure 2.** Phylogenetic position of Nuosu proper.

```
Sichuan Yi (or Nuosu proper)
        •       Niesu
                •       Adur
                •       Suondi
        •       Nuosu
                •       Shynra
                •       Yynuo
                •       Qumusu
```

**Figure 3.** Internal subgroupings of Nuosu proper.

A close dialect of Adur Niesu is Nuosu. Nuosu, also meaning 'black people', is a relatively well-studied variety of Nuosu proper (Chen et al. 1985; Bradley 1990; Chen and Da 1998; Lama 1998; Hu 2001, 2010; Gerner 2013). Both Niesu and Nuosu are classified under Nuosu proper (Lama 2012); see Figure 3. People using the autonym of Nuosu include Shynra, Yynuo, and Qumusu speakers, whose population is estimated to be about 1.9 million (Bradley 2001).

The mutual intelligibility between Adur Niesu and Nuosu is relatively low (Bradley 2001), which is mainly due to phonological differences (see Table 1, and also Pan 2001; Matisoff 2006; Hajek 2006; Lama 2012).

While Adur shares many words with Suondi Niesu, it is phonologically different from Nuosu. If their geographic distribution is considered, Suondi Niesu is sandwiched between Adur Niesu and Nuosu. According to Lama (2022), there are two shared phonological innovations in Adur Niesu and Suondi Niesu, making them different from Nuosu. The first one is the lenition of the voiceless nasals, namely, making the voiceless nasals m̥ and n̥ voiced; see Table 2. The second innovation is that the *o sound in Proto-Nuosu proper is fronted and raised to i in Adur and Suondi Niesu; see Table 2.

**Table 1.** Exemplifying the phonological differences between Niesu and Nuosu.

| | Adur Niesu | Suondi Niesu | Shynra Nuosu |
|---|---|---|---|
| 'arrive' | tɕʰi³³ | ɕi³³ | ɕi³³ |
| 'die' | ʂi³³ | si³³ | si³³ |
| 'dog' | tʂʰi³³ | tsʰi³³ | kʰɯ³³ |
| 'half' | tɕɛ³³pʰjɛ³³ | tɕɛ³³pʰjɛ³³ | tɕɛ³³pʰa³³ |
| 'head' | o³³tɕʰɯ³³ | o³³tɕʰi³³ | i³³tɕʰi³³ |
| 'mate' | bu⁴⁴dzɯ³³ | bo⁴⁴dzɯ³³ | bo⁴⁴dzɯ³³ |
| 'nose' | na²¹bi⁵⁵ | na²¹bi⁵⁵ | na̠²¹bi⁵⁵ |
| 'reciprocal, together' | dzɨ³³ | dzɨ³³ | dʑi³³ |
| 'first half of the month' | zɨ³³ | zɨ³³ | zi³³ |
| 'see' | ŋɯ²¹ | ŋɯ²¹/ hɯ²¹ | hɯ²¹ |
| 'take' | ɕi²¹ | si²¹ | si²¹ |
| 'second half of the month' | djɛ³³ | djɛ³³ | do³³ |
| 'waist, middle' | dʑo⁵⁵ | dʑo⁵⁵ | dʑu⁵⁵ |
| autonym | njɛ³³su³³ | njɛ³³su³³ | nɔ³³su³³ |

The identical words are highlighted among Adur Niesu, Suondi Niesu and Shynra Nuosu.

**Table 2.** Niesu phonological innovations (Lama 2022).

| Proto-Nuosu Proper | Shynra Nuosu | Suondi Niesu | Adur Niesu | Meaning |
|---|---|---|---|---|
| *m̥a⁵⁵mo²¹ | m̥a⁵⁵mo²¹ | ma⁵⁵mo²¹ | ma⁵⁵mo²¹ | 'teacher' |
| *m̥o⁵⁵ | m̥o⁵⁵ | mi⁵⁵ | mi⁵⁵ | 'soldier' |
| *a³³n̥i³³ | a³³n̥i³³ | a³³ni³³ | a³³ni³³ | 'red' |
| *n̥a²¹bi⁵⁵ | n̥a²¹bi⁵⁵ | na²¹bi⁵⁵ | na²¹bi⁵⁵ | 'nose' |
| *tʰo⁵⁵ | tʰo⁵⁵ | tʰi⁵⁵ | tʰi⁵⁵ | 'upper part' |
| *vo⁵⁵ | vo⁵⁵ | vi⁵⁵ | vi⁵⁵ | 'pig' |
| *lo⁵⁵ | lo⁵⁵ | li⁵⁵ | li⁵⁵ | 'hand' |
| *zo⁵⁵ | zo⁵⁵ | zi⁵⁵ | zi⁵⁵ | 'to entertain' |

There are additional innovations to subgroup Adur Niesu and Suondi Niesu under one node, and support Lama's (2022) claim that they should be the first group to branch off from Proto-Nuosu proper. For example, the Proto-Loloish (PL) stops in Table 3 change to affricates in Adur and Suondi Niesu. It is also interesting to observe an intermediate stage towards affrication in the Jiaojihe (literally 'intercourse river' or 交际河) variety of Adur Niesu, which is to the south of Butuo and adjacent to the northeastern border of Yunnan. In Jiaojihe variety, the velar plosive is kept and the fricative is epenthesized. This could be considered as a shift of place of articulation from velar plosive to retroflex affricate, and is probably a feature of Proto-Adur Niesu.

**Table 3.** Examples of affrication in Niesu.

| | Shynra Nuosu | Suondi Niesu | Adur Niesu (Jiaojihe) | Adur Niesu (Tuojue) | PL/PLB/PTB |
|---|---|---|---|---|---|
| 'dog' | kʰɯ³³ | tsʰi³³ | kʰʂi³³ | tʂʰi³³ | *kwe² |
| 'bird nest' | kʰɯ³³ | tsʰi³³ | kʰʂi³³ | tʂʰi³³ | *kʷəy¹ |
| 'to untie' | pʰu³³ | tsʰi³³ | kʰʂi³³ | tʂʰi³³ | *pre¹ |
| 'evening' | kʰɯ⁵⁵ | tsʰi⁵⁵ | kʰʂi⁵⁵ | tʂʰi⁵⁵ | *ʔ-kutᴸ |
| 'sun' | gɯ³³ | dzi³³ | gzi³³ | dʑi³³ | *m-ka-n |

The reconstruction *kwe² (PL), *pre¹ (PL), and *ʔ-kutᴸ (PL) are taken from Bradley (1979); *kʷəy¹ (PLB) and *m-ka-n (PTB) from Matisoff (2003).

Another innovation is the insertion of a medial /w/ to form diphthongs after velars (see Matisoff 2006; Bradley 2008). See examples in Table 4. The diphthongization is still

stable in Adur Niesu. According to Lama (2022), the diphthongization, however, is being lost among young Suondi Niesu speakers, while this feature has still been kept among the elder Suondi speakers.

**Table 4.** Examples of diphthongization in Niesu (Lama 2022, with revision).

| Proto-Nuosu Proper | Shynra Nuosu | Suondi Niesu | Adur Niesu | Meaning |
|---|---|---|---|---|
| *gwo$^{33}$ | bo$^{33}$ | gwi$^{33}$ | gwi$^{33}$ | 'to go' |
| *gi$^{55}$ | gi$^{55}$ | gwi$^{55}$ | gwi$^{55}$ | 'be childless' |
| *k$^{h}$e$^{33}$ | k$^{h}$e$^{33}$ | k$^{h}$we$^{33}$ | k$^{h}$wɛ$^{33}$ | 'to chop' |
| *ŋge$^{33}$ | ŋge$^{33}$ | ŋgwe$^{33}$ | ŋgwɛ$^{33}$ | 'to lie, cheat' |
| *k$^{h}$a$^{55}$ | k$^{h}$a$^{55}$ | k$^{h}$wa$^{55}$ | k$^{h}$wa$^{55}$ | 'be happy' |

Moreover, velars in Nuosu are more palatalized in Suondi Niesu and Adur Niesu, if followed by the front vowel /i/ (see Table 5).

**Table 5.** Examples of palatalization in Adur Niesu.

| | Shynra Nuosu | Suondi Niesu | Adur Niesu |
|---|---|---|---|
| 'to jolt or winnow (e.g., grain)' | k$^{h}$i$^{55}$ | tɕi$^{33}$ | tɕi$^{33}$ |
| 'to ladle; scoop out with a spoon' | k$^{h}$i$^{55}$ | tɕ$^{h}$i$^{55}$ | tɕ$^{h}$i$^{55}$ |
| 'spade hoe, a three-spiked digging tool' | (lɔ$^{55}$)gɔ$^{21}$ | gɔ$^{33}$ | (la$^{55}$)dʑi$^{55}$ |
| 'to put the roof on (a thatched house)' | ki$^{55}$ | - | tɕ$^{h}$i$^{55}$ |

There are also phonological features which make Adur Niesu distinctive from Suondi Niesu. Table 3 shows that Adur Niesu retroflexizes the alveolar affricates in Suondi Niesu. The retroflexization, as a typical feature of Adur Niesu, is the reflex of PL or PTB *r and PL *ʃor *s; see Table 6.

**Table 6.** Examples of retroflexes in Adur Niesu.

| | Shynra Nuosu | Suondi Niesu | Adur Niesu (Jiaojihe) | Adur Niesu (Tuojue) | PL/PTB |
|---|---|---|---|---|---|
| 'gallbladder' | tɕɿ$^{33}$ | - | kʂɿ$^{33}$ | ʈʂɿ$^{33}$ | *m/s-kri(y)-s |
| 'copper' | dʑɿ$^{33}$ | dzɿ$^{33}$ | gzɿ$^{33}$ | dʐɿ$^{33}$ | *gre$^{2}$ |
| 'skin' | ndʑɿ$^{33}$ | ndzɿ$^{33}$ | ŋgzɿ$^{33}$ | nḍʐɿ$^{33}$ | *re$^{1}$ |
| 'big' | ʑɿ$^{33}$ | zɿ$^{33}$ | zɿ$^{33}$ | ʐɿ$^{33}$ | *k/ʔ-ri$^{2}$ |
| 'foot' | ɕɿ$^{33}$ | sɿ$^{33}$ | ʂɿ$^{33}$ | ʂɿ$^{33}$ | *kre$^{1}$ |
| 'to die' | sɿ$^{33}$ | sɿ$^{33}$ | ʂɿ$^{33}$ | ʂɿ$^{33}$ | *ʃe$^{2}$ |
| nominalizer | su$^{33}$ | ʂu$^{33}$ | ʂu$^{33}$ | ʂu$^{33}$ | *su$^{1}$ |

All reconstructions are taken from PL in Bradley (1979), except *m/s-kri(y)-s (PTB) from Matisoff (2003).

Vowel-wise, the front vowel /i/ in Suondi Niesu, as well as Nuosu, corresponds to back vowel /ɯ/ in Adur Niesu if they are preceded by alveolo–palatal sounds (see Table 7).

**Table 7.** Examples of correspondence to back vowel /ɯ/ in Adur Niesu.

| | Shynra Nuosu | Suondi Niesu | Adur Niesu (Tuojue) |
|---|---|---|---|
| 'to become' | dʑi$^{21}$ | dʑi$^{21}$ | dʑɯ$^{21}$ |
| 'bee' | dʑi$^{33}$ | dʑɿ$^{33}$ | dʑɯ$^{33}$ |
| 'leaf' | tɕ$^{h}$i$^{33}$ | tɕ$^{h}$i$^{33}$ | tɕ$^{h}$ɯ$^{33}$ |
| 'egg' | tɕ$^{h}$i$^{21}$ | tɕ$^{h}$i$^{21}$ | tɕ$^{h}$ɯ$^{21}$ |
| 'to precipitate (e.g., rain)' | dʑi$^{21}$ | dʑi$^{21}$ | dʑɯ$^{21}$ |

### 3. Segmental Phonology

This section starts with Adur Niesu consonants and then moves on to vowels. After introducing the syllable and the phonotactics, segmental changes in both vowels and consonants will be covered.

### *3.1. Consonants*

Table 8 demonstrates the 41 phonemic consonants of Adur Niesu: nine plain plosives, three prenasalized plosives, eleven fricatives, four nasals, two laterals, nine affricates and three prenasalized affricates. Suondi Niesu has the same consonant inventory as Adur Niesu (Lama 2012; Mise 2020). Compared with Nuosu, Adur Niesu lacks voiceless nasals /m̥/ and /n̥/ (see Section 2). Depending on whether a consonant can precede either the unrounded palatal [j] or the rounded labiovelar [w], Adur Niesu consonants can be divided into two groups: the J-group, marked in the solid box, and the W-group, marked in the dotted box. The other consonants cannot be followed by the glides.

**Table 8.** Adur Niesu consonants.

|  | **Bilabial** | **Dental** | **Retroflex** | **Alveolo–Palatal** | **Velar** | **Glottal** |
|---|---|---|---|---|---|---|
| Plosive | mb | nd |  |  | ŋg |  |
|  | b | d |  |  | g |  |
|  | p | t |  |  | k |  |
|  | pʰ | tʰ |  |  | kʰ |  |
| Fricative | v | z | ʐ | ʑ | ɣ |  |
|  | f | s | ʂ | ɕ | x | h |
| Affricate |  | ndz | ndʐ | ndʑ |  |  |
|  |  | dz | dʐ | dʑ |  |  |
|  |  | ts | tʂ | tɕ |  |  |
|  |  | tsʰ | tʂʰ | tɕʰ |  |  |
| Nasal | m | n |  | ɳ | ŋ |  |
| Lateral |  | l |  |  |  |  |
|  |  | ɬ |  |  |  |  |

#### 3.1.1. Plain Plosives

The plain plosives are differentiated from the prenasalized plosives (see Section 3.1.5). They are produced through three places of articulation: bilabial, dental, and velar, as shown in Table 9, respectively. The three-way contrast among the plain plosives is achieved with voiced vs. voiceless unaspirated vs. voiceless aspirated. While the velar group cannot go with the J-glide, the bilabial and dental groups cannot go with the W-glide. It should be noted that the diphthongs [jɛ] and [wɛ] are two allophones of /ɛ/ and [wi] is an allophone of /i/ (see Section 3.2.1).

**Table 9.** Adur Niesu plosives.

| **Bilabial** | **Dental** | **Velar** |
|---|---|---|
| b | d | g |
| p | t | k |
| pʰ | tʰ | kʰ |

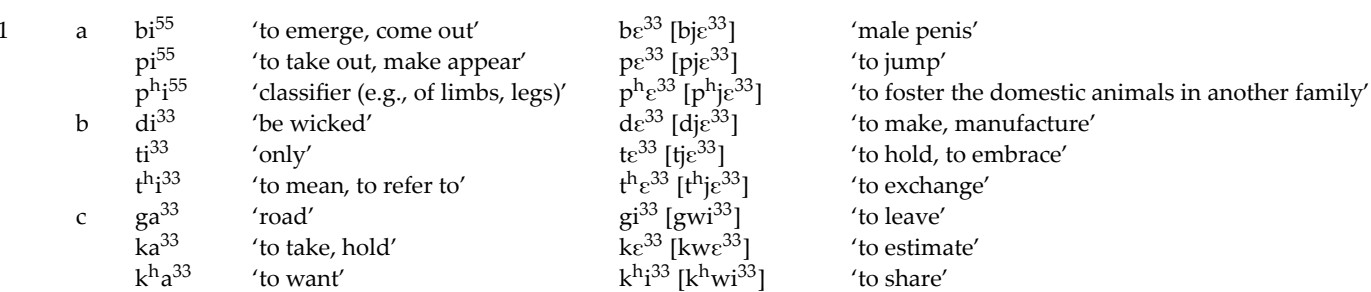

| 1 | a | bi⁵⁵ | 'to emerge, come out' | bɛ³³ [bjɛ³³] | 'male penis' |
|---|---|---|---|---|---|
|  |  | pi⁵⁵ | 'to take out, make appear' | pɛ³³ [pjɛ³³] | 'to jump' |
|  |  | pʰi⁵⁵ | 'classifier (e.g., of limbs, legs)' | pʰɛ³³ [pʰjɛ³³] | 'to foster the domestic animals in another family' |
|  | b | di³³ | 'be wicked' | dɛ³³ [djɛ³³] | 'to make, manufacture' |
|  |  | ti³³ | 'only' | tɛ³³ [tjɛ³³] | 'to hold, to embrace' |
|  |  | tʰi³³ | 'to mean, to refer to' | tʰɛ³³ [tʰjɛ³³] | 'to exchange' |
|  | c | ga³³ | 'road' | gi³³ [gwi³³] | 'to leave' |
|  |  | ka³³ | 'to take, hold' | kɛ³³ [kwɛ³³] | 'to estimate' |
|  |  | kʰa³³ | 'to want' | kʰi³³ [kʰwi³³] | 'to share' |

### 3.1.2. Fricatives

The eleven fricatives are articulated at six places: bilabial, dental, retroflex, alveolo–palatal, velar and glottal (see Table 10).

**Table 10.** Adur Niesu fricatives.

| Bilabial | Dental | Retroflex | Alveolo–Palatal | Velar | Glottal |
|----------|--------|-----------|-----------------|-------|---------|
| v | z | ʐ | ʑ | ɣ | |
| f | s | ʂ | ɕ | x | h |

At each place, except glottal, the fricative pair contrasts in terms of voicing. The five pairs of fricatives are exemplified in the following minimal pairs. No fricatives can go with a glide, neither the J-glide nor the W-glide.

| 2 | a | $vi^{55}$ | 'pig' | $fi^{55}$ | 'cliff, stomach' |
|---|---|-----------|-------|-----------|------------------|
| | b | $zi^{55}$ | 'to pay off, to unload' | $si^{55}$ | 'to kill' |
| | c | $ʐo^{33}$ | 'sheep' | $ɕo^{33}$ | 'to grow, to raise' |
| | c | $za^{33}$ | 'to shout' | $ʂɯ^{33}$ | 'be toilsome' |
| | d | $ɣo^{33}$ | 'blue, green' | $xo^{33}$ | 'a ring (for keeping animals or for crematorium)' |
| | e | - | | $ho^{33}$ | 'to marry' |

### 3.1.3. Nasals and Laterals

The nasals have four places of articulation: bilabial, dental, alveolo–palatal and velar, and the laterals have just one: dental (see Table 11).

**Table 11.** Adur Niesu nasals and laterals.

| Bilabial | Dental | Alveolo–Palatal | Velar |
|----------|--------|-----------------|-------|
| m | n | ɲ | ŋ |
| | l | | |
| | ɬ | | |

Unlike Nuosu, the Niesu bilabial and dental nasals do not have a voicing contrast. They can go with the J-glide. The velar nasal can go with the W-glide.

| 3 | a | $mi^{33}$ | 'name' | b | $ma^{55}$ | 'to teach' | c | $mɛ^{33}$ $[mjɛ^{33}]$ | 'to lift with feet (as in wrestling)' |
|---|---|-----------|--------|---|-----------|------------|---|------------------------|----------------------------------------|
| | | $ni^{33}$ | 'female' | | $na^{55}$ | 'to coax' | | $nɛ^{33}$ $[njɛ^{33}]$ | 'black' |
| | | - | | | $ŋa^{55}$ | 'to install' | | $ŋi^{33}$ $[ŋwi^{33}]$ | 'front' |
| | | $ɲi^{33}$ | 'also' | | $ɲa^{55}$ | 'be late' | | | |

There is a contrast between voiced /l/ and voiceless /ɬ/. When the laterals precede the front vowel /ɛ/, they may be palatalized, showing free variation.

| 4 | a | $li^{55}$ | 'to lick' | b | $lɛ^{33}$ | 'a verbal classifier (e.g., amount of the effort)' | $[lɛ^{33}]$ or $[ljɛ^{33}]$ |
|---|---|-----------|-----------|---|-----------|-----------------------------------------------------|------------------------------|
| | | $ɬi^{55}$ | 'be young' | | $ɬɛ^{33}$ | 'to extract' | $[ɬɛ^{33}]$ or $[ɬjɛ^{33}]$ |

### 3.1.4. Affricates

Niesu, both Adur and Suondi, has three sets of affricates, produced at dental, retroflex and alveolo–palatal, respectively. Each set shows a three-way contrast in terms of voicing and aspiration, as exemplified below (see Table 12).

| 5 | a | $dzi^{33}$ | 'be left over' | $tsi^{33}$ | 'to leave something behind' | $ts^{h}i^{33}$ | 'to fall quickly' |
|---|---|------------|----------------|------------|------------------------------|-----------------|--------------------|
| | b | $dʐɯ^{33}$ | 'money' | $tʂɯ^{33}$ | 'to feed, make eat' | $tʂ^{h}ɯ^{33}$ | 'rice, crop' |
| | c | $dʑi^{33}$ | 'to fall down' | $tɕi^{33}$ | 'a classifier (e.g., clothes)' | $tɕ^{h}i^{33}$ | 'to arrive' |

**Table 12.** Adur Niesu affricates.

| Dental | Retroflex | Alveolo–Palatal |
|---|---|---|
| dz | ɖʐ | dʑ |
| ts | ʈʂ | tɕ |
| tsʰ | ʈʂʰ | tɕʰ |

### 3.1.5. Prenasalized Consonants

Voiced plosives and affricates are prenasalized in Adur Niesu (see Table 13). The prenasalized consonants are treated here as unitary segments, not consonant clusters, on the ground that (1) they are contrastive with other consonants, such as ndo²¹ 'to fall down' vs. do²¹ 'speech, word', ŋga³³ 'be clever' vs. ga³³ 'road', and nɖʐa³³ 'to sprinkle water for cooking the corn rice' vs. ɖʐa³³ 'sparrow'; (2) the nasal is always homorganic with the following plosives or affricates; and (3) the nasal–obstruent onsets only appear in the syllable-initial position. Lama (1998) also considers prenasalized obstruents in Nuosu, a close dialect of Adur Niesu, unitary segments, not consonant clusters, after acoustic analysis.

**Table 13.** Adur Niesu prenasalized consonants.

| Labial | Dental | Retroflex | Alveolo–Palatal | Velar |
|---|---|---|---|---|
| mb | nd | | | ŋg |
| | ndz | nɖʐ | ndʑ | |

The prenasalized plosives can also be followed by the glides.

| 6 | a | mbi³³ | 'to divide' | mbɛ³³ [mbjɛ³³] | 'to shoot' |
|---|---|---|---|---|---|
| | b | ndo³³ | 'to drink' | ndɛ³³ [ndjɛ³³] | 'a kind of black bowl painted with lacquer tree liquid' |
| | c | ŋga³³ | 'be clever' | ŋgi³³ [ŋgwi³³] | 'to chew' |
| | d | ndza⁵⁵ | 'adjacency' | | |
| | e | nɖʐa⁵⁵ | 'be good' | | |
| | f | ndʑi⁵⁵ | 'to catch' | | |

### 3.1.6. Glides

Adur Niesu distinguishes between two glides: the unrounded palatal j and the rounded labiovelar w. The former is non-phonemic and the latter is phonemic. The glides are treated as part of the rhyme of a syllable, but not an element in a complex consonant. The reason is based on economy. By doing this, the sum of the diphthongs formed by the two glides is only four, including three allophonic diphthongs, [wɛ], [wi] and [jɛ] (see Section 3.2.1), and one phonemic diphthong, /wa/ (see Section 3.2.4), exemplified below. Bradley (2008) treated the glide /w/ as an element in complex consonants, or labialized velars. However, if similar treatment is made to the glide j, there would be as many as 17 complex consonants, including 13 allophonic complex consonants, such as [bʲ], [pʲ], [pʰʲ], [dʲ], [mʲ], and [lʲ] and four phonemic ones: gʷ, kʷ, kʰʷ and ŋʷ. This greatly exceeds the sum of the diphthongs formed by the two glides.

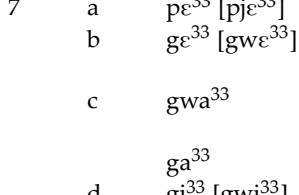

| 7 | a | pɛ³³ [pjɛ³³] | 'to jump' | tʰɛ³³ [tʰjɛ³³] | 'to exchange' |
|---|---|---|---|---|---|
| | b | gɛ³³ [gwɛ³³] | 'to compete with speaking skills' | kʰɛ³³ [kʰwɛ³³] | 'to chop' |
| | c | gwa³³ | 'be of high capacity' | kʰwa³³ | 'to share excessive important livestock, e.g., female pig, cattle (but need to pay back)' |
| | | ga³³ | 'road' | kʰa³³ | 'to want' |
| | d | gi³³ [gwi³³] | 'to leave, to go' | kʰi³³ [kʰwi³³] | 'to share' |

### 3.2. Vowels

There are 10 monophthongs and one diphthong in Adur Niesu: /i/, /ɨ/, /ɯ/, /o/, /u/, /ɛ/, /ʝ/, /a/, /ɔ/, /u̠/, and /wa/. Lama (2012), Mahai (2015, 2019) and Mise (2020) reported

similar monophthongs in Suondi Niesu. But different symbols are used, namely, the /ɿ ʅ/ set in Mise (2020) is represented as /i ɨ/ in the present study, and /e/ in Lama (2012) as /ɛ/ in the present study. All Adur vowels are oral. The monophthongs are organized by height (high, mid, low) and backness (front, central, back). A feature of Adur Niesu vowels is high vowel fricativization, occurring with the two high central vowels, /i/ and /ɨ/, and the two high back vowels /u/ and /u̠/.

It should be noted that the Adur vowel /ɯ/ is more advanced and lower than the cardinal IPA [ɯ]. Due to this deviation, it is not impossible to transcribe this vowel as /ə/, such as in the Nuosu vowel inventory in Lama (2002). In the present study, /ɯ/ is used, mainly because the Adur Niesu /ɯ/ is categorically closer to the cardinal IPA [ɯ] in terms of vowel height and backness. This symbol /ɯ/ is also adopted in describing Nuosu vowels (Lama 1998; Edmondson et al. 2017).

Another way of organizing Niesu vowels is to categorize them into tense and lax vowels (Chen et al. 1985; Lama 2002) (see Table 14). This is useful for the description of vowel assimilation (see Section 3.4.2). It should be noted that the tense/lax contrast in the tradition of Southeast Asian languages have been applied in reversed fashion to the terms that are used in talking about Germanic languages (Maddieson and Ladefoged 1985). The principal component of the tense/lax distinction in Adur Niesu, as well as other Yi languages, is a difference in the laryngeal setting, namely, the tense vowels are more laryngealized than the lax ones (Lama 2002; Esling and Edmondson 2002). Therefore, the lax vowels are closer in the vowel space and, thus, higher, while the tense vowels are more open and, thus, lower (Edmondson et al. 2017). Therefore, Adur Niesu monophthongs can be paired as below. This pairing also displays frequent assimilation results discussed in Section 3.4.

**Table 14.** Tense/lax pairs of monophthongs in the Adur dialect.

| Monophthongs | Lax vowel | i | ɨ | ɯ | o | u |
| | Tense vowel | ɛ | ɨ̠ | a | ɔ | u̠ |

### 3.2.1. Front Vowels

The Adur Niesu front vowels are distinguished by height. The minimal pairs are below.

| 8 | a | fi³³ | 'to throw' | fɛ³³ | 'mouse' | fa³³ | 'golden pheasant' |
| | b | dzi³³ | 'be left over' | dzɛ³³ | 'to carve' | dza³³ | 'rice, food' |
| | c | tsʰi³³ | 'ten' | tsʰɛ³³ | 'deer' | tsʰa³³ | 'be hot' |
| | d | tɕʰi³³ | 'to arrive' | tɕʰɛ³³ | 'to jump' | tɕʰa³³ | 'to caw (e.g., crow)' |

The vowel /i/ cannot follow the retroflexes and the velar fricatives. It has an allophone [wi] when it occurs with velar stops.

| 9 | a | ŋgi³³ | [ŋgwi³³] | 'to chew' |
| | b | gi³³ | [gwi³³] | 'to leave' |
| | c | ki³³ | [kwi³³] | 'to dare' |
| | d | kʰi³³ | [kʰwi³³] | 'to share' |
| | e | ŋi³³ | [ŋwi³³] | 'front' |

The vowel /ɛ/ has two diphthong allophones, [jɛ] and [wɛ]. The diphthong [jɛ] occurs when it follows the J-group consonants and the diphthong [wɛ] occurs when it follows the W-group consonants of velar.

| 10 | a | mbɛ³³ | [mbjɛ³³] | 'to shoot' | gɛ³³ | [gwɛ³³] | 'to compete with speaking skills' |
| | b | mɛ³³ | [mjɛ³³] | 'to lift with feet' | kʰɛ³³ | [kʰwɛ³³] | 'to chop' |
| | c | ɬɛ³³ | [ɬjɛ³³] | 'to extract' | (o⁵⁵)ŋɛ³³ | [ŋwɛ³³] | 'be hungry' |
| | d | nɛ³³ | [njɛ³³] | 'black' | kɛ³³ | [kwɛ³³] | 'to estimate' |

### 3.2.2. Central Vowels

Adur central vowels contrast one another in terms of height. The contrast between ɨ and ɨ̠, regarding the retractedness or tenseness, also exists in Nuosu (Edmondson et al. 2017).

| 11 | a | pɨ³³ | 'to exhibit speaking skills' | pɨ̠³³ | 'be not generous' |
|----|---|------|------------------------------|------|-------------------|
|    | b | pʰɨ³³ | 'be painful, be spicy' | pʰɨ̠³³ | CLF (for farmland) |
|    | c | zɨ³³ | 'to buy' | zɨ̠³³ | 'to press' |
|    | d | ʂɨ³³ | 'to die' | ʂɨ̠³³ | 'to roar' |

Vowels ɨ and ɨ̠ only occur with 17 consonants: the three plain bilabial plosives, the six dental fricatives and affricates, the six retroflex fricatives and affricates, and the two dental laterals. Both of them are subject to high vowel fricativization, each having two allophones in the form of fricative vowels, namely, [z̩] and [z̠̩], when they follow the plosives and the dentals, and [ʐ̩] and [ʐ̠̩] when they follow the retroflex sounds. Therefore, the phonetic realizations of the examples in (11) are (12a) to (12d). See more examples in (12e) to (12m). According to Edmondson et al. (2017, p. 89), Nuosu expresses 'dragon' with the fricative vowel [v] as an allophone of /u/, thus transcribed phonetically with labialization: lʷ̩³³ 'dragon' (cf. lu³³ as the phonemic form). However, in Adur Niesu, lip rounding is not observed in the pronunciation of 'dragon' (see 12l) or in the other examples transcribed with labialization in Edmondson et al. (2017). Therefore, the laterals are incompatible with /u/ and /u̠/ in Adur Niesu. Similar to Nuosu, the laterals in Adur Niesu will end up being rhoticized after the high vowel fricativization. Similar rhoticization is reported in Ersu, a Na–Qiangic language spoken in Liangshan (Chirkova and Handel 2013).

| 12 | a | pɨ | [pz̩³³] | 'to exhibit speaking skills' | pɨ̠³³ | [pz̠̩³³] | 'be not generous' |
|----|---|------|---------|------------------------------|------|---------|-------------------|
|    | b | pʰɨ³³ | [pʰz̩³³] | 'be painful, spicy' | pʰɨ̠³³ | [pʰz̠̩³³] | CLF (for farmland) |
|    | c | zɨ³³ | [zz̩³³] | 'to buy' | zɨ̠³³ | [zz̠̩³³] | 'to press' |
|    | d | ʂɨ³³ | [ʂʐ̩³³] | 'to die' | ʂɨ̠³³ | [ʂʐ̠̩³³] | 'to roar' |
|    | e | bɨ | [bz̩³³] | 'to give' | (zɨ³³)bɨ̠³³ | [bz̠̩³³] | 'a bamboo trap' |
|    | f | sɨ³³ | [sz̩³³] | 'again, still' | sɨ̠³³ | [sz̠̩³³] | 'tree' |
|    | g | dzɨ³³ | [dzz̩³³] | 'to ride (horse)' | dzɨ̠³³ | [dzz̠̩³³] | CLF (for vehicles) |
|    | h | tsʰɨ³³ | [tsʰz̩³³] | 'he, she, it' | tsʰɨ̠³³ | [tsʰz̠̩³³] | 'generation' |
|    | i | ʐɨ³³ | [ʐʐ̩³³] | 'be big' | ʐɨ̠³³ | [ʐʐ̠̩³³] | 'soul' |
|    | j | dʐɨ³³ | [dʐʐ̩³³] | 'teeth' | dʐɨ̠³³ | [dʐʐ̠̩³³] | 'to exist' |
|    | k | tʂɨ³³ | [tʂʐ̩³³] | 'to squeeze oil' | tʂɨ̠³³ | [tʂʐ̠̩³³] | 'to pull' |
|    | l | lɨ³³ | [lz̩³³] | 'dragon' | lɨ̠³³ | [lz̠̩³³] | 'to shake' |
|    | m | ɬɨ³³ | [ɬz̩³³] | 'wind' | ɬɨ̠³³ | [ɬz̠̩³³] | 'to escape' |

The rounded vowel /o/ may be reduced to [ə] by some Adur Niesu speakers. Other than this, large-scale patterned vowel centralization is not found in Adur Niesu.

| 13 | | a⁴⁴ɳo³³ | 'many' | [a⁴⁴ɳo³³] | or | [a⁴⁴ɳə³³] |
|----|---|---------|--------|-----------|----|-----------|

### 3.2.3. Back Vowels

Similar to /ɨ/ and /ɨ̠/, the vowel /u/ contrasts with /u̠/ in terms of retractedness. Although the Adur vowel /ɯ/ is more advanced and lower than the cardinal IPA [ɯ], it is discussed with other back vowels.

| 14 | a | pʰu³³ | 'price, value' | pʰu̠³³ | 'to dig' | - | |
|----|---|-------|----------------|--------|----------|------|---|
|    | b | ʂu³³ | 'to do' | - | | ʂɯ³³ | 'to find' |
|    | c | ŋgu³³ | 'to boast' | ŋgu̠³³ | 'to rub with hands' | ŋgɯ³³ | 'buckwheat' |

The phonemic contrast between /u/ and /u̠/ is not symmetric. Consonants that can occur with /u/ may not have a contrast with /u̠/. For instance, zu³³ 'to irritate' has no contrastive pair as *zu̠³³, and kʰu³³ 'to steal' lacks a contrast as *kʰu̠³³. It is, therefore, observed that /u/ and /u̠/ start to merge as one phoneme. According to the main consultants, the following pairs are interchangeable, showing free variations.

| 15 | a | su̱³³ga⁵⁵ | 'be rich' | su̱³³ga⁵⁵ | or | su̱³³ga⁵⁵ |
|----|---|-----------|-----------|-----------|-----|-----------|
|    | b | bu̱⁵⁵ | 'grass' | bu̱⁵⁵ | or | bu⁵⁵ |
|    | c | gu̱⁵⁵ | 'to sew the clothes' | gu̱⁵⁵ | or | gu⁵⁵ |
|    | d | ku̱⁵⁵ | 'to know how' | ku̱⁵⁵ | or | ku⁵⁵ |
|    | e | vu̱⁵⁵ | 'to run over' | vu̱⁵⁵ | or | vu⁵⁵ |
|    | f | fu̱⁵⁵ | 'to work hard' | fu̱⁵⁵ | or | fu⁵⁵ |
|    | g | mu̱⁵⁵ | 'to sip' | mu̱⁵⁵ | or | mu⁵⁵ |
|    | h | ŋgu̱⁵⁵ | 'to stab' | ŋgu̱⁵⁵ | or | ŋgu⁵⁵ |

Another observation is that the retracted /u̱/ is forming a complementary distribution with /u/ by occurring with the high-level tone 55 only. While the lax /u/ bears tone 33, the tense /u̱/ bears tone 55 in (16). Mise (2020) also indicates that tone 55 causes vowel tenseness in Suondi Niesu. Therefore, it is possible for the two phonemes to merge or become allophonic in the future.

| 16 | a | du̱³³ | 'wing' | *du̱³³ | du̱⁵⁵ | 'be stealthy' |
|----|---|--------|--------|--------|--------|---------------|
|    | b | ŋu³³ | 'be' | *ŋu̱³³ | (kɔ³³lɔ³³)ŋu̱⁵⁵ | 'be angry' |
|    | c | tu³³ | 'to lift' | *tu̱³³ | tu̱⁵⁵(m̩³³) | 'be promising' |

Both /u/ and /u̱/ are noteworthy in that they lead to syllabic consonants if they are preceded by /m/. It was clearly observed from the consultants that the two syllables below were not produced with any rounding of the lips.

| 17 | mu³³ | [m̩³³] | 'to do' | mu̱³³ | [m̱̩³³] | 'to blow up' |
|----|------|--------|---------|--------|---------|--------------|

Due to high vowel fricativization, like /ɨ/ and /ɨ̱/, /u/ and /u̱/ have an allophone of their own in the form of the fricative vowel [v̩] and [v̱̩] when they are preceded by velar consonants. It was observed from the consultants that the upper teeth touched the inner side of the lower lip when they pronounced these syllables, without any rounding of the lips.

| 18 | a | ŋgu³³ | [ŋgv̩³³] | 'to boast' | ŋgu̱³³ | [ŋgv̱̩³³] | 'to rub with hands' |
|----|---|--------|-----------|------------|---------|-----------|---------------------|
|    | b | gu³³ | [gv̩³³] | 'be firm' | gu̱³³ | [gv̱̩³³] | 'to protect (food, cubs)' |
|    | c | ku³³ | [kv̩³³] | 'to call' | ku̱⁵⁵ | [kv̱̩⁵⁵] | 'to know how' |
|    | d | kʰu³³ | [kʰv̩³³] | 'to steal' | - | | |

A final feature of /u/ and /u̱/ is that they may be substituted with a syllabic bilabial trill [ʙ] after labial and dental plosives. The trill substitution is subject to personal habit, thus forming free variation. But the trill substitution is more preferred after voiced labial and dental plosives, and less preferred after voiceless ones.

| | | /u/ | | | /u̱/ | |
|----|---|-----|---|---|------|---|
| 19 | a | [bu²¹] or [bʙ²¹] | 'a clan' | [bu̱³³] or [bʙ̱³³] | 'to write' | |
|    | b | [pu²¹] or [pʙ²¹] | 'to carry' | - | | |
|    | c | [pʰu³³] or [pʰʙ³³] | 'value' | [pʰu̱³³] or [pʰʙ̱³³] | 'to dig' | |
|    | d | [du³³] or [dʙ³³] | 'wing' | [du̱⁵⁵] or [dʙ̱⁵⁵] | 'be stealthy' | |
|    | e | [tu³³] or [tʙ³³] | 'to lift' | - | | |
|    | f | [tʰu³³] or [tʰʙ³³] | 'silver' | [tʰu̱³³ (or tʰʙ̱³³) ʂa³³] | 'a kind of evil spirit' | |
|    | g | [mbu³³] or [mbʙ³³] | 'be trapped' | [mbu̱³³] or [mbʙ̱³³] | 'to squint' | |
|    | h | [ndu²¹] or [ndʙ²¹] | 'to beat' | [tjɛ³³ndu̱³³ (or ndʙ̱³³)] | 'be fat' | |

The mid-high oral vowel /o/ forms a phonemic contrast with the mid-low one /ɔ/ in terms of the openness of the mouth.

| 20 | a | bo³³ | 'mountain' | bɔ³³ | 'to demand' |
|----|---|------|------------|------|-------------|
|    | b | pʰo³³ | 'to run' | pʰɔ³³ | 'to reverse' |
|    | c | ko³³ | 'side, location' | kɔ³³ | 'be capable' |

### 3.2.4. Diphthong

While no diphthong is reported in the Nuosu vowel inventory (Lama 1998, 2002; Gerner 2013; Edmondson et al. 2017), different numbers of diphthongs in Suondi Niesu are reported: /ie ui ue/ in Lama (2012), /uɑ ue ui/ in Mahai (2015) and /ua ui/ in Mise (2020).

Due to the close relation between Suondi and Adur Niesu, it is suspected that not all reported diphthongs in Suondi Niesu are phonemic.

In Adur Niesu, phonetically, there are four diphthongs, [jɛ], [wɛ], [wi] and [wa]. But the only phonemic diphthong in Adur Niesu is /wa/. It can only occur with velar consonants, or the W-group. Minimal pairs are as shown below.

| 21 | a | gwa$^{33}$ | 'be of large capacity' | | ga$^{33}$ | 'to wear' |
|---|---|---|---|---|---|---|
| | b | kwa$^{33}$ | 'fire pit' | | ka$^{33}$ | 'to take' |
| | c | k$^{h}$wa$^{33}$ | 'to share excessive important livestock, e.g., female pig, cattle (but need to pay back)' | | k$^{h}$a$^{33}$ | 'to want' |
| | d | ŋwa$^{55}$ | 'to break apart' | | ŋa$^{55}$ | 'to install' |

### 3.3. The Syllable and Phonotactics

Adur Niesu syllable structure is relatively simple. All are open syllables. Adur Niesu segments are organized into syllables as below (see Figure 4).

| Onset | Rhyme | |
|---|---|---|
| (Consonant) | (On-glide) | vowel or syllabic consonant |

**Figure 4.** Syllable structure of Adur Niesu.

The onset can be any of the 41 consonants. The on-glides are either j or w. The vowel slot can be filled by any of the 10 monophthongs or the syllabic consonants if there is no glide in the syllable. The J-glide only occurs with /ɛ/, and the W-glide occurs with all front vowels, namely, /i/, /ɛ/ and /a/. But syllables involving a glide must be preceded by an onset, such as (22a) to (22f). In this case, all slots are filled. The onset and on-glide slots can be optional; see (22g) to (22h). The following are examples of all possible syllables in Adur Niesu. Most Adur Niesu syllables are made up of a consonant and a vowel.

| 22 | a | ŋgwi$^{33}$ | 'to chew' |
|---|---|---|---|
| | b | kwi$^{33}$ | 'to dare' |
| | c | mbjɛ$^{33}$ | 'to shoot' |
| | d | gwɛ$^{33}$ | 'to break' |
| | e | k$^{h}$wa$^{55}$ | 'be happy' |
| | f | ŋwa$^{55}$ | 'to break apart' |
| | g | a$^{44}$mo$^{33}$ | 'mother' |
| | h | o$^{33}$ | 'head' |

Without considering the three basic tones of Adur Niesu, there are 308 attested syllables. Allophonic realizations are indicated in Table 15.

**Table 15.** Adur Niesu phonotactics.

| | ɨ | ɨ̠ | i | ɛ | ɯ | a | o | ɔ | u | u̠ | wa | |
|---|---|---|---|---|---|---|---|---|---|---|---|---|
| zero | - | - | + | + | - | + | + | + | - | - | - | 5 |
| b | z̩ | z̠ | + | jɛ | - | + | + | + | + | + | - | 9 |
| mb | - | - | + | jɛ | - | + | + | + | + | + | - | 7 |
| p | z̩ | z̠ | + | jɛ | - | + | + | + | + | - | - | 8 |
| pʰ | z̩ | z̠ | + | jɛ | - | + | + | + | + | + | - | 9 |
| d | - | - | + | jɛ | + | + | + | + | + | + | - | 8 |
| nd | - | - | + | jɛ | + | + | + | + | + | + | - | 8 |
| t | - | - | + | jɛ | + | + | + | + | + | + | - | 8 |
| tʰ | - | - | + | jɛ | + | + | + | + | + | - | - | 7 |
| m | - | - | + | jɛ | - | + | + | + | m̩ | m̠ | - | 7 |
| n | - | - | + | jɛ | + | + | + | + | - | - | - | 6 |
| l | z̩ | z̠ | + | jɛ | + | + | + | + | - | - | - | 8 |
| ɬ | z̩ | z̠ | + | jɛ | + | + | + | + | - | - | - | 8 |
| g | - | - | wi | wɛ | + | + | + | + | v̩ | v̠ | + | 9 |
| ŋg | - | - | wi | wɛ | + | + | + | + | v̩ | v̠ | - | 8 |
| k | - | - | wi | wɛ | + | + | + | + | v̩ | v̠ | + | 9 |
| kʰ | - | - | wi | wɛ | + | + | + | + | v̩ | v̠ | + | 9 |
| ŋ | - | - | wi | wɛ | + | + | + | + | + | + | + | 9 |
| v | - | - | + | + | - | + | + | + | + | + | - | 7 |
| f | - | - | + | + | - | + | + | - | + | + | - | 6 |
| z | z̩ | z̠ | + | + | + | + | + | + | + | - | - | 9 |
| s | z̩ | z̠ | + | + | + | + | + | + | + | + | - | 10 |
| ʐ | ʑ̩ | ʑ̠ | - | + | + | + | + | + | + | - | - | 8 |
| ʂ | z̩ | ʑ̠ | - | + | + | + | + | + | + | - | - | 8 |
| ʑ | - | - | + | + | + | + | + | + | + | - | - | 7 |
| ç | - | - | + | + | - | - | + | - | - | - | - | 3 |
| ɣ | - | - | - | - | + | + | + | + | - | - | - | 4 |
| x | - | - | - | - | + | + | + | + | - | - | - | 4 |
| h | - | - | + | + | - | + | + | + | - | - | - | 5 |
| dz | z̩ | z̠ | + | + | + | + | + | + | + | - | - | 9 |
| ndz | z̩ | z̠ | + | + | + | + | + | + | + | - | - | 9 |
| ts | z̩ | z̠ | + | + | + | + | + | + | + | - | - | 9 |
| tsʰ | z̩ | z̠ | + | + | + | + | + | + | + | - | - | 9 |
| dʐ | ʑ̩ | ʑ̠ | - | - | + | + | + | + | + | - | - | 7 |
| ndʐ | ʑ̩ | ʑ̠ | - | + | + | + | + | + | + | - | - | 8 |
| tʂ | ʑ̩ | ʑ̠ | - | + | + | + | + | + | + | - | - | 8 |
| tʂʰ | ʑ̩ | ʑ̠ | - | + | + | + | + | + | + | - | - | 8 |
| dʑ | - | - | + | + | + | + | + | + | - | - | - | 6 |
| ndʑ | - | - | + | + | - | + | + | + | - | - | - | 5 |
| tɕ | - | - | + | + | + | + | + | + | - | - | - | 6 |
| tɕʰ | - | - | + | + | + | + | + | + | - | - | - | 6 |
| ɲ | - | - | + | + | - | + | + | + | - | - | - | 5 |
| | ɨ | ɨ̠ | i | ɛ | ɯ | a | o | ɔ | u | u̠ | wa | 308 |

## 3.4. Segmental Changes in Vowels

In the previous sections, some vowel changes were discussed: allophones of the front vowels in Section 3.2.1, occasional vowel reduction in Section 3.2.2, and high vowel fricativization in Sections 3.2.2 and 3.2.3. In the present section, another four vowel changes are presented: vowel lowering, vowel centralization, vowel assimilation and vowel fusion.

### 3.4.1. Vowel Lowering and Centralization

The high vowel /u/ may be lowered to /o/, forming a free variation. The reason why the change is considered a lowering, rather than a raising, is that the high vowel /u/ is more common in the speech of both the elder and young population.

| 23 | | a | su⁴⁴zɨ³³ | 'the elder' | [su⁴⁴]~[so⁴⁴] |
|----|--|---|----------|-------------|----------------|
| | | b | ʐu³³ | 'to catch' | [ʐu³³]~[ʐo³³] |

It is common for the Adur Niesu back vowel /u/ to be centralized as /ɨ/ if it follows the sibilant fricatives.

| 24 | nɛ³³ | 'black' | + | su³³ | 'people' | → | nɛ³³su³³ or | nɛ³³sɨ³³ | 'Niesu people or Niesu language' |
|----|------|---------|---|------|----------|---|-----------|---------|----------------------------------|
| | a²¹ | | + | su⁵⁵ | | → | **a²¹su⁵⁵** or | **a²¹sɨ⁵⁵** | 'we (inclusive)' |
| | zo³³ | 'study' | + | ʂu³³ | nominalizer | → | zo³³ʂu³³ or | zo³³ʂɨ³³ | 'those (who are) studying' |

### 3.4.2. Vowel Assimilation

Vowel assimilation is another case of vowel lowering in Adur Niesu. Nearly all assimilations in Adur Niesu are regressive, and most occur between tense and lax vowels (see Section 3.2), namely, the preceding lax vowel will be lowered to a tense vowel, or become more laryngealized; see Table 14. Recall in Section 3.2 that the tense vowels are treated as those which are more laryngealized than the lax ones, and thus have a lower position than the lax ones (Maddieson and Ladefoged 1985; Lama 2002; Edmondson et al. 2017). Therefore, the rhyme of the first syllable is assimilated in terms of the tenseness of the following rhyme. Compare the examples in (25a) to (25d). /ɛ/, /a/ and /ɨ/, belonging to the tense group, lower the vowel of the first syllable from the lax one to its tense counterpart, namely, from [o] to [ɔ] and from [i] to [ɛ], respectively. But if the following rhymes do not belong to the tense group, assimilation does not occur.

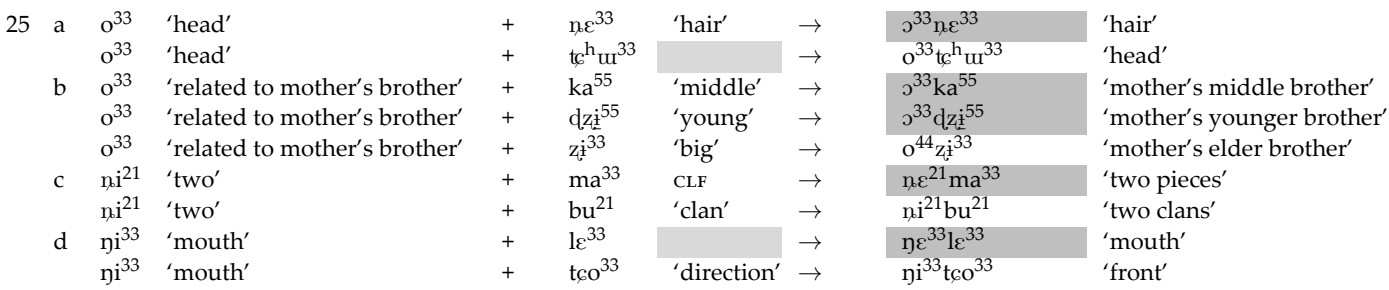

| 25 | a | o³³ | 'head' | + | n̠ɛ³³ | 'hair' | → | ɔ³³n̠ɛ³³ | 'hair' |
|----|---|------|--------|---|-------|--------|---|----------|--------|
| | | o³³ | 'head' | + | tɕʰɯ³³ | | → | o³³tɕʰɯ³³ | 'head' |
| | b | o³³ | 'related to mother's brother' | + | ka⁵⁵ | 'middle' | → | ɔ³³ka⁵⁵ | 'mother's middle brother' |
| | | o³³ | 'related to mother's brother' | + | dʑɨ⁵⁵ | 'young' | → | ɔ³³dʑɨ⁵⁵ | 'mother's younger brother' |
| | | o³³ | 'related to mother's brother' | + | zɨ³³ | 'big' | → | o⁴⁴zɨ³³ | 'mother's elder brother' |
| | c | n̠i²¹ | 'two' | + | ma³³ | CLF | → | n̠ɛ²¹ma³³ | 'two pieces' |
| | | n̠i²¹ | 'two' | + | bu²¹ | 'clan' | → | n̠i²¹bu²¹ | 'two clans' |
| | d | ŋi³³ | 'mouth' | + | lɛ³³ | | → | ŋɛ³³lɛ³³ | 'mouth' |
| | | ŋi³³ | 'mouth' | + | tɕo³³ | 'direction' | → | ŋi³³tɕo³³ | 'front' |

More examples are as below.

| 26 | a | zɨ³³ | 'water' | + | tsɨ³³ | 'to get drenched' | → | zɨ³³tsɨ³³ | 'to get drenched by rain' |
|----|---|------|---------|---|-------|-------------------|---|-----------|---------------------------|
| | b | mu³³ | 'ground' | + | lɨ³³ | 'to shake' | → | mu⁴⁴lɨ³³ | 'earthquake' |
| | c | o³³ | 'head' | + | ma³³ | general classifier | → | ɔ³³ma³³ | 'head, mind' |
| | d | ni³³ | 'female' | + | ndʐa⁵⁵ | 'be good, pretty' | → | nɛ³³ndʐa⁵⁵ | 'beautiful woman' |
| | e | do²¹ | 'speech' | + | ma³³ | general classifier | → | dɔ²¹ma³³ | 'speech, word' |
| | f | zɯ³³ | 'son, man' | + | kʰɔ³³ | 'be capable' | → | za³³kʰɔ³³ | 'hero, brave man' |
| | g | ʐo³³ | 'sheep' | + | la³³ | 'breeding' | → | ʐɔ³³la³³ | 'ram that is not castrated' |
| | h | zɯ³³ | 'son, man' | + | tɕʰɛ³³ | 'hard' | → | za³³tɕʰɛ³³ | 'tough man' |
| | i | ni³³ | 'red' | + | tʂɨ³³tʂɨ³³ | 'IDE' | → | nɛ⁴⁴tʂɨ³³tʂɨ³³ | 'bright red' |
| | j | ni³³ ndʐa⁵⁵ | | + | ni³³ vɛ³³ | | → | nɛ³³ndʐa⁵⁵nɛ³³vɛ³³ | 'beautiful woman' |

Some of the assimilations are more phonetic in nature, since they can be restored to the original vowel in slow and careful speech; but some are more morpholexical in nature, since they cannot be restored to the original vowel, even using slow and careful speech. If restoration is forced in the latter case, new meanings will be produced. All examples in (27) are phonetic assimilations and those in (28) are morpholexical assimilations.

| 27 | | *casual speech* | *careful speech* |
|----|--|-----------------|-------------------|
| | 'tough man' | za³³kʰɔ³³ | zɯ³³kʰɔ³³ |
| | 'brightly red' | nɛ⁴⁴tʂɨ³³tʂɨ³³ | ni⁴⁴tʂɨ³³tʂɨ³³ |
| | 'see' | ɣo²¹ŋo³³ | ɣɯ²¹ŋo³³ |

28　　*With assimilation*　　　　　　　　　　　　　　*Without assimilation*

　　dɔ²¹ma³³　　　'speech, language'　　　do²¹ma³³　　　'one piece of speech'
　　ʐo³³la³³　　　'ram'　　　　　　　　　　ʐo³³la³³　　　'the sheep come'
　　za³³tɕʰɛ³³　　'capable man'　　　　　　zɯ³³tɕʰɛ³³　　'the man is capable'
　　nɛ³³ndʐa⁵⁵　　'beautiful woman'　　　　ni³³ndʐa⁵⁵　　'the woman is beautiful'
　　ɔ³³tʂɨ⁵⁵　　　'plait'　　　　　　　　　o³³tʂɨ⁵⁵　　　'to plait'
　　ɔ³³ma³³　　　'head, mind'　　　　　　o³³ma³³　　　'a head'

　　　　　　　The tenseness/laxness-induced assimilation can also be relative. As long as the fol-
　　　　lowing vowel is tenser, or lower, than the preceding vowel, regressive assimilation can be
　　　　triggered. For example, although /o/ is laxer than /ɔ/, it is tenser and lower than /ɯ/; there-
　　　　fore, assimilation occurs in (29). Since /u/ is not tenser than /i/, assimilation is not triggered
　　　　in ȵi²¹bu²¹ 'two households' in (25c) (cf. ȵɛ²¹ma³³ 'two pieces').

29　　ɣɯ²¹　　'to obtain'　　　　　+　　　ŋo³³　　'to see'　　→　　ɣo²¹ŋo³³　　'see'

### 3.4.3. Vowel Fusion

　　　　　　Vowel fusion in Adur Niesu results in vowel substitution of the rhyme of the preced-
　　　　ing syllable, such as ʑa³³ (ʑi³³ 'to go' + a³³ 'attitudinal marker') 'let's go'. Although it is
　　　　also possible for vowel fusion to occur intraclausally, it is more common at the clause end.
　　　　In (30) and (31), the rhyme of the first syllable is replaced by the following vowel at the
　　　　clause's final position, and in (32), the vowel fusion occurs in the clause.

30　la³³=**si³³**　**o⁴⁴**　　　　　→　　　　　　　　la³³=**so⁴⁴**
　　come=REP PFV　　　　　　　　　　　　　come=REP.PFV
　　'(He) came again.'

31　ʂu³³ + o⁴⁴ → ʂo⁴⁴
　　lɯ³³　　　　tsʰɨ³³　　　tɕi³³　　ʂa³³mu³³ȵi³³　　a³³tsʰɨ³³~tsʰɨ³³　mu³³　　ndu²¹　　ʂɨ²¹=**ʂɔ⁴⁴**.
　　cow　　　　this　　　CLF　　toilsome:do:EXST　extreme~REDPL　do　　beat　　die=NMLZ.ATT
　　'This cow was beaten to death pathetically.'

32　tsʰɨ³³ + a²¹=si²¹ → tsʰa²¹=si²¹
　　ŋo³³　　　　**tsʰa²¹**=si²¹
　　1PL　　　　3SG.NEG=know
　　'We do not know it.'

### 3.5. Segmental Changes in Consonants

　　　　　　Segmental changes in Adur Niesu consonants are not widely observed. Lenition and
　　　　clanlects are presented.

### 3.5.1. Lenition of the Velar Consonants

　　　　　　Briefly, the velar stops can be lenited in spontaneous speech as velar fricatives; see (33).

33　lenition　　　ŋo³³ 'we'　　　→　　　　ɣo³³
　　lenition　　　ko³³ 'when, if'　→　　　　xo³³
　　lenition　　　ga⁴⁴dʑɨ³³　　　→　　　　ɣa⁴⁴dʑɨ³³
　　　　　　　　'absolutely'

### 3.5.2. Aspiration of the Clanlects

　　　　　　Variations in aspiration change between different clans are found, or 'clanlects'. One
　　　　of the main consultants is a descendant of the dʑɛ²¹nɛ³³ clan, and another one is of the
　　　　su̠³³ga⁵⁵ clan. Both of them live in the same village since they were born. The following
　　　　two words are not aspirated in the speech of the dʑɛ²¹nɛ³³ descendant, while they are
　　　　aspirated in the speech of the su̠³³ga⁵⁵ descendant. But the aspirated affricate in the three
　　　　words has wider usage among Adur Niesu speakers. Other than the two words, both
　　　　consultants share similar phonological system of Adur Niesu.

|   |   |   |  | dzɛ²¹nɛ³³ clan | su̠³³ga⁵⁵ clan |
|---|---|---|---|---|---|
| 34 | a | 'he, she, it' |  | tsɿ³³ | tsʰɿ³³ |
|   | b | 'this' |  | tsɿ³³ | tsʰɿ³³ |

### 3.6. Syllable Reduction

It is common for Adur Niesu syllables to be reduced in continuous speech. There are three types of syllable reduction being observed in the field: complete reduction including the segment and tone, partial reduction with a floating tone left, and partial reduction with the initial consonant left.

The syllable is so reduced that a particle is left to signal the existence of a clause; see (35b) where tsʰɿ²¹mu³³ 'doing so' is reduced.

| 35 | a. | **tsʰɿ²¹** | **mu³³** | ta³³, | | hɛ³³ŋga⁵⁵ | lɯ³³o³³=sa⁴⁴ | lɯ³³ʂɿ³³ ŋgɔ³³ | la³³. |
|---|---|---|---|---|---|---|---|---|---|
|   |   | this | do | NF | | Han Chinese | cow:head=COM | cow:foot collect | come |

'By doing so, the Han Chinese come to collect the head and feet of the (killed) cow.'

| | b. | ta³³, | he³³ŋga⁵⁵ | lɯ³³o³³=sa⁴⁴ | | lɯ³³ʂɿ³³ | ŋgɔ³³ | la³³. |
|---|---|---|---|---|---|---|---|---|
| | | NF | Han Chinese | cow:head=COM | | cow:foot | collect | come |

'By doing so, the Han Chinese come to collect the head and feet of the (killed) cow.'

It is often the segments of the whole syllable being deleted. After the reduction, the tone becomes a floating tone and reassociates itself onto the preceding syllable. For example, in (36a) to (36c), the second syllable is reduced, namely, ʂɿ in a³³ʂɿ⁵⁵ 'what', sɿ in a²¹sɿ²¹tʰɯ³³ 'when' and hi in a²¹hi³³ 'cannot'. But the tone is left. The tonal trace can be observed on the remaining preceding syllable. Namely, the original tone of the preceding syllable is overridden by the floating tone, where a³³ changes to a⁵⁵ in (36a) and a²¹ changes to a³³ in (36c). Since the first syllable a²¹ bears the same 21 tone with the deleted syllable in (36b), the overriding is not evident. In (36a) and (36b), other than the syllable reduction, the fricative glottal /h/ can often be epenthesized, namely, ha⁵⁵ and ha²¹.

| 36 | a | a³³ʂɿ⁵⁵ | tsɿ⁵⁵ | → | **a⁵⁵ / ha⁵⁵** | | tsɿ⁵⁵ |
|---|---|---|---|---|---|---|---|
|   |   | what | do | | what | | do |

'to do what'

| | b | a²¹sɿ²¹tʰɯ³³ | | → | **a²¹tʰɯ³³ / ha²¹tʰɯ³³** |
|---|---|---|---|---|---|
| | | 'which:time' | | | 'which:time' |
| | | 'when' | | | |

| | c | a²¹hi³³ mu³³ (NEG=can do) 'cannot' → **a³³mu³³** |
|---|---|---|

| | | tɯ²¹ | ʑi³³ | **a³³** | mu³³ | | tʰɯ³³ | dzɿ³³ |
|---|---|---|---|---|---|---|---|---|
| | | rise | go | NEG.CAN | do | | place | EXST |

'(Someone) cannot stand up, but keep staying there.'

In a polar interrogative, on the surface, there seems to be a tone change: 55 > 21 / 55 _. However, the tone lowering from 55 to 21 is not a tone change (cf. tone sandhi in Section 4.2 and Section 4.3), but in fact the result of the floating tone associated with the interrogative particle a²¹ after syllable reduction, which is exemplified below. The floating tone of the interrogative particle overrides the tone of the preceding syllable. Meanwhile, the preceding high front vowel [i] is assimilated by the interrogative particle a²¹ and lowered to [ɛ] (see Section 3.4). If the lowered vowel [ɛ] occurs with the J-group consonants, it will subsequently change to the phonetic diphthong allophone [jɛ], namely, pɛ²¹ [pjɛ²¹] in (37b) and ndɛ²¹ [ndjɛ²¹] in (37d).

| | | basic form | meaning | reduplicated form + interrogative particle | result | meaning |
|---|---|---|---|---|---|---|
| 37 | a | si⁵⁵ | 'to kill' | si⁵⁵~si⁵⁵ + a²¹ | si⁵⁵~sɛ²¹ | 'to kill or not' |
|   | b | pi⁵⁵ | 'to dig' | pi⁵⁵~pi⁵⁵ + a²¹ | pi⁵⁵~pɛ²¹ | 'to dig or not' |
|   | c | vi⁵⁵ | 'to shoulder' | vi⁵⁵~vi⁵⁵ + a²¹ | vi⁵⁵~vɛ²¹ | 'to shoulder or not' |
|   | d | ma²¹ma²¹ ndi⁵⁵ | 'to bear fruit' | ma²¹ma²¹ ndi⁵⁵~ndi⁵⁵ + a²¹ | ma²¹ma²¹ ndi⁵⁵~ndɛ²¹ | 'to bear fruit or not' |
|   | e | ndzɿ³³ ʑi⁵⁵ | 'to be drunk' | ndzɿ³³ ʑi⁵⁵~ʑi⁵⁵ + a²¹ | ndzɿ³³ ʑi⁵⁵~ʑɛ²¹ | 'to be drunk or not' |

It is particularly useful to contrast the above syllable reduction with reduplication for intensification in Adur Niesu. Without the effect of the interrogative particle, when two high-level tones are adjacent to each other, there is no change of the tone and of the vowel.

| | | basic form | meaning | reduplicated form | meaning |
|---|---|---|---|---|---|
| 38 | a | si$^{55}$ | 'to kill' | si$^{55}$~si$^{55}$ | 'to kill fiercely' |
| | b | ma$^{21}$ma$^{21}$ ndi$^{55}$ | 'to bear fruit' | ma$^{21}$ma$^{21}$ ndi$^{55}$~ndi$^{55}$ | 'to bear a lot of fruits' |
| | c | ndzɨ$^{33}$ ʐi$^{55}$ | 'to be drunk' | ndzɨ$^{33}$ ʐi$^{55}$~ʐi$^{55}$ | 'to be quite drunk' |

Moreover, the syllable reduction also occurs to other vowels bearing the high-level tone 55; see (39a) to (39c), accompanied by vowel assimilation. However, the syllable reduction does not occur to syllables bearing other non-high-level tones; see (39d) and (39e). Likewise, the vowel assimilation will not occur.

| | | basic form | meaning | reduplicated form | meaning |
|---|---|---|---|---|---|
| 39 | a | tʂɨ$^{55}$ | 'be correct' | tʂɨ$^{55}$~tʂɨ$^{55}$ + a$^{21}$ or tʂɨ$^{55}$~tʂɨ$^{21}$ | 'be correct or not' |
| | b | sa$^{55}$ | 'to finish' | sa$^{55}$~sa$^{55}$ + a$^{21}$ or sa$^{55}$~sa$^{21}$ | 'to finish or not' |
| | c | pʰo$^{55}$ | 'to dig the earth' | pʰo$^{55}$~pʰo$^{55}$ + a$^{21}$ or pʰo$^{55}$~pʰɔ$^{21}$ | 'to dig the earth or not' |
| | d | pʰi$^{33}$ | 'be polite' | pʰi$^{44}$~pʰi$^{33}$ + a$^{21}$ | 'be polite or not' |
| | e | fi$^{33}$ | 'to throw' | fi$^{44}$~fi$^{33}$ + a$^{21}$ | 'to throw or not' |

Sometimes, the syllable may not be completely reduced, leaving not only the tone, but also the onset. The leftover will go with the preceding syllable; see (40). Mahai (2019) reports another kind of partial reduction, namely, the initial consonant is deleted, with only the rhyme left, such as ʑa$^{21}$o$^{55}$ for ʑa$^{21}$ʑo$^{55}$ 'potato' and ŋo$^{21}$i$^{55}$ for ŋo$^{21}$ɲi$^{55}$ 'the two of us (exclusive)'. However, we did not have a similar observation about this reduction in Adur Niesu in the field.

40  a$^{21}$ŋu$^{33}$ 'now' → aŋ$^{33}$

| aŋ$^{33}$ | tsʰi$^{44}$~tsʰi$^{33}$=ko$^{44}$=na$^{33}$, | | li$^{55}$sa$^{33}$ | tsʰi$^{21}$ | ma$^{33}$ | ka$^{33}$ |
|---|---|---|---|---|---|---|
| present | neat~neat=moment=DSC | | official seal | one | CLF | take |
| a$^{33}$tu̱$^{33}$ | ɔ$^{21}$la$^{33}$mu$^{33}$hi$^{55}$ bɨ$^{33}$ | xa$^{33}$ | ʑi$^{33}$ | o$^{33}$ | di$^{44}$. | |
| Adur | name | give | away | go down | PFV | QUOT |

'(Someone) said that at this exact moment, (you) took an official seal and went to give (it) to Uolamuhi who is from the Adur region.'

Syllable reduction can also create the environment for vowel assimilation. For example, tsʰi$^{21}$ mu$^{33}$ ɔ$^{44}$nɔ$^{33}$ (this do if) 'if it is like this' changes to tsʰɔ$^{33}$ɔ$^{44}$nɔ$^{33}$, with the syllable mu$^{33}$ being deleted, namely, tsʰi$^{33}$ ɔ$^{44}$nɔ$^{33}$, and tone 33 being reassociated to the preceding syllable and the rhyme then being assimilated by the following tenser vowel ɔ.

| 41 | a | o$^{21}$, | **tsʰi$^{21}$** | **mu$^{33}$** | **ɔ$^{44}$nɔ$^{33}$,** | ŋo$^{33}$ | tsʰi$^{33}$ | a$^{21}$=si$^{21}$. |
|---|---|---|---|---|---|---|---|---|
| | | INTJ | this | do | if | 1PL.EXCL | 3SG | NEG=know |

'Oh, if it is something like this, we do not know it.'

| | b | o$^{21}$, | **tsʰɔ$^{33}$ɔ$^{44}$nɔ$^{33}$,** | ŋo$^{33}$ | tsʰi$^{33}$ | a$^{21}$=si$^{21}$. |
|---|---|---|---|---|---|---|
| | | INTJ | this | 1PL.EXCL | 3SG | NEG=know |

'Oh, if it is something like this, we do not know it.'

## 4. The Suprasegmentals

Adur Niesu employs suprasegmentals as an important means for lexical contrast, like many other syllable–tone languages of East and Southeast Asia. In Adur Niesu, two types of tonal alternation should be distinguished: tone sandhi and tone change. Similar distinction is made in Prinmi (Ding 2014) and in Yongning Na (or Narua) (Michaud 2017).

Tone sandhi refers to the phonologically conditioned tonal alternation by adjacent tones, regardless of the morphosyntactic factors. The most productive sandhi rule of Adur Niesu is 33 > 44 / _ 33, such as su$^{33}$ 'people' + zɨ$^{33}$ 'big' > su$^{44}$zɨ$^{33}$ 'the elder'.

Tone change is governed by rules that are confined to specific morphosyntactic environments. It is the dominant form of tonal alternation in Adur Niesu. The tone change ap-

pears in the following morphosyntactic contexts: (1) compound words, (2) prefixed words, (3) patient marking, and (4) yes–no interrogation generated by reduplication.

Finally, floating tones in Adur Niesu can generate a surface kind of tonal alternation, although, in fact, it is the result of syllable reduction. After the syllable reduction, the tone becomes a floating tone and reassociates itself onto the preceding syllable, such as the so-called tonal change regarding the possessive pronouns, where the tone of the reduced genitive marker *ni$^{21}$ of Proto-Nuosu proper was retained by the plain personal pronouns in Adur Niesu.

### 4.1. The Three Basic Tones

Identical to Suondi Niesu, Adur Niesu has three basic tones: high-level tone 55, mid-level tone 33, and low-falling tone 21. The minimal contrast between these three tonal categories is exemplified below (see Figure 5).

42   a   di$^{21}$   'to say'     di$^{33}$   'be not good'     di$^{55}$   'to wear (shoes)'
    b   ti$^{21}$   'to bury'     ti$^{33}$   'be only'     ti$^{55}$   'to make wear (shoes)'
    c   vi$^{21}$   'guest'     vi$^{33}$   possessive pronominal enclitic     vi$^{55}$   'pig'
    d   hi$^{21}$   'to say'     hi$^{33}$   'house'     hi$^{55}$   'eight'
    e   ts$^{h}$i$^{21}$   'his, her, its'     ts$^{h}$i$^{33}$   'he, she, it'     ts$^{h}$i$^{55}$   'family line'
    f   to$^{21}$   'can'     to$^{33}$   'to respond'     to$^{55}$   'to light up'

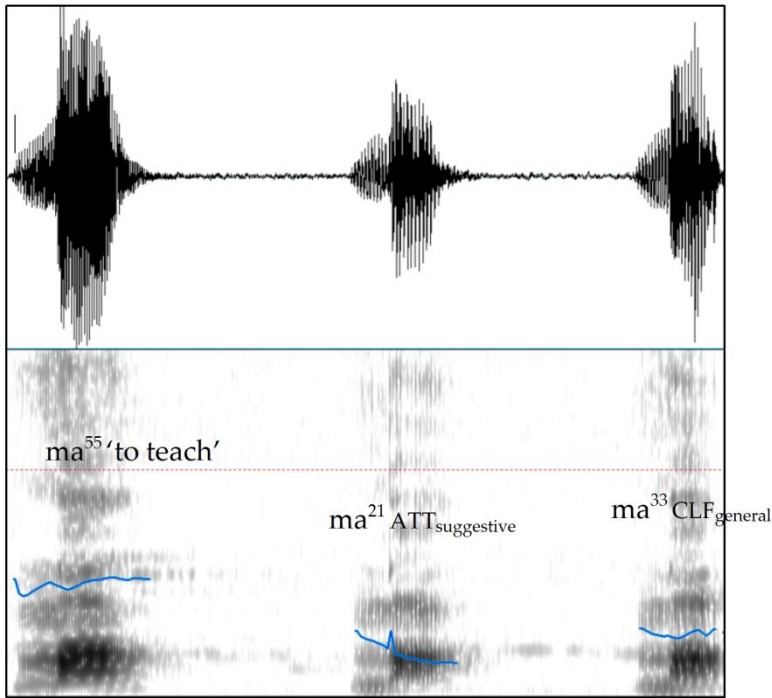

**Figure 5.** Adur Niesu tones exemplified by syllable [ma].

There is a 44 (high-mid level) tone in Adur Niesu. See Bradley (1990) for the discussion of tone 44 in Nuosu. However, it is seen largely in cases of tone sandhi, which often results from either tone 33 or tone 21 in syllable combination. There is no co-occurrence of tone 44 with tone 55 at the lexical level. In Figure 6, tone 44 is slightly higher than tone 33 in the word pi$^{33}$mo$^{44}$ 'priest', but tone 55 is much higher than tone 33 in the word nɛ$^{33}$ndʐa$^{55}$ 'pretty woman'.

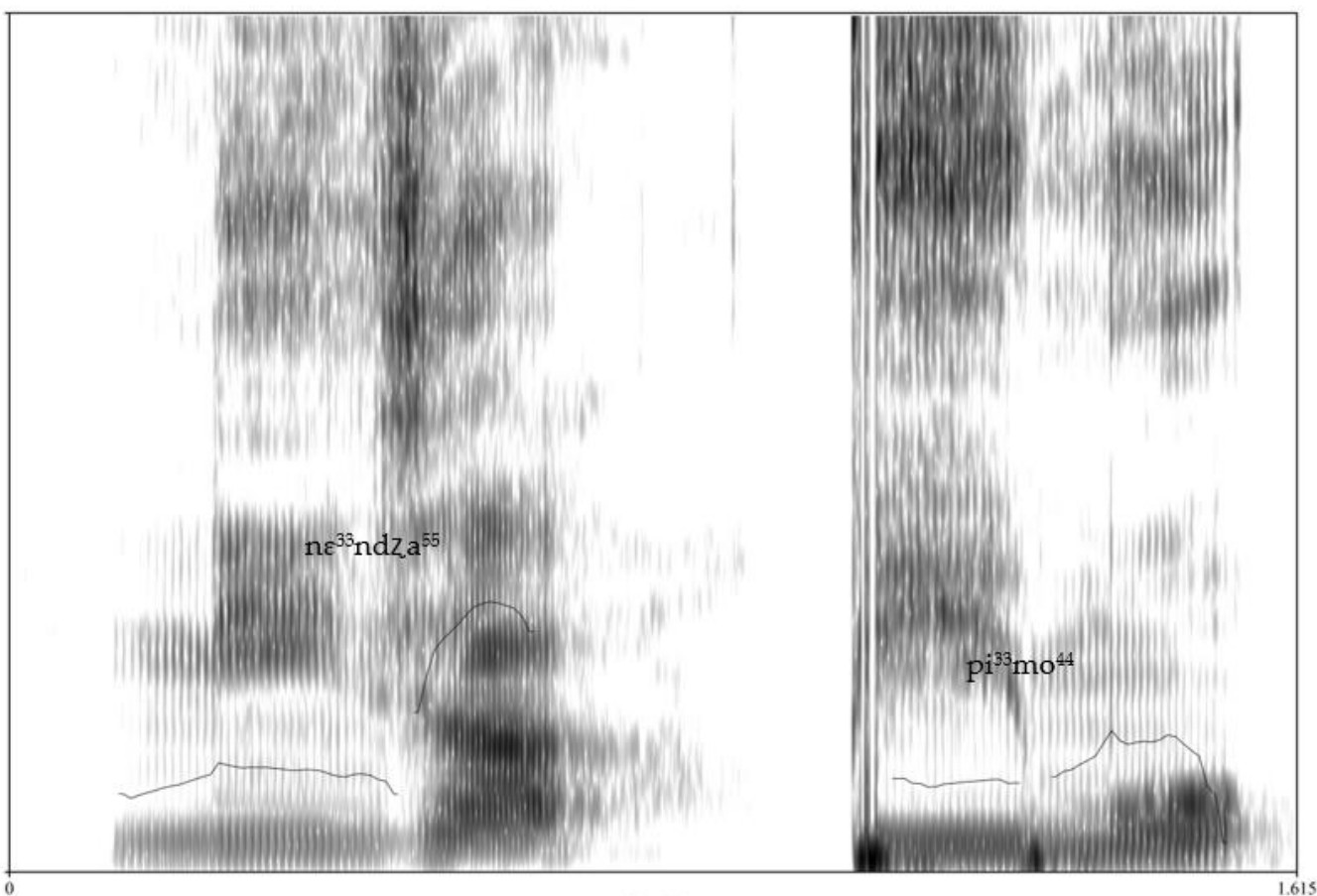

**Figure 6.** Compare Adur Niesu tone 44 with tone 33 and tone 55.

Tone 44 often appears in particles at the clause boundary, such as the sequential clitic ɕi⁴⁴ and change of state clitic o⁴⁴ in (43), and clause linker lɯ⁴⁴ in (44). If the clause boundary is occupied by content words, tone 44 is not used, such as lɨ³³ 'to trap' in (44). If not used at the clause boundary, tone 44 only appears in a few morphemes in Adur Niesu as citation forms, namely, mo⁴⁴ as a hesitator, sa⁴⁴ the comitative, di⁴⁴ the quotative, and ŋo⁴⁴ the experiential clitic.

| 43 | no²¹ | a²¹mu³³ | tsʰɨ³³ | ŋɛ²¹ | ma³³ |
|---|---|---|---|---|---|
| | 2PL.POSS | daughter | this | two | CLF |
| | tʂʰɨ⁵⁵=**ɕi⁴⁴**, | o³³no³³ | xa³³ | a⁴⁴nɛ³³, | |
| | tie=SEQ | distance | release | after | |
| | no²¹ | zɯ³³ | ma⁴⁴ | sa³³ | **o⁴⁴**. |
| | 2PL.POSS | son | CLF.DEF | comfortable | CSM |

'After (you) tie up your two daughters and abandon them in the wilderness, your son will recover.'

| 44 | to⁵⁵ | pa²¹nɛ²¹=ko³³ | lɨ³³, | ga²¹mo²¹=ko³³ | tsi⁴⁴ | | tɯ³³ | **lɯ⁴⁴**, |
|---|---|---|---|---|---|---|---|---|
| | stamp | mud=LOC | trap | road:big=LOC | place inside | | CONT | CLNK |
| | bi⁵⁵la³³ | a³³=to²¹ mu³³ | | tʰɯ³³ | i⁵⁵ | dʐɨ³³ | ta³³, . . . . | |
| | exit:come | NEG=can do | | t/here | lie | CONT | NF | |

'(The horse, the bull and all the big beasts) stamped on (the frog) into the mud, (who was) being stuck firmly in the road2, and (the frog) could not come out, staying there, . . . .'

### 4.2. Tone Sandhi: 33 > 44 / _ 33

The most productive tone sandhi in Adur Niesu is 33 > 44 / _ 33, regardless of the morphosyntactic environment. Other phonological processes may also occur, such as vowel

assimilation in (45f) and (45g). The fundamental function of this tone sandhi is to dissimilate two adjacent same tone.

| 45 | a | su$^{33}$ 'people' | + | zɨ$^{33}$ 'big' | → | su$^{44}$zɨ$^{33}$ | 'the elder' |
|----|---|----|---|----|---|----|----|
| | b | a$^{33}$ | + | nɛ$^{33}$ | → | a$^{44}$nɛ$^{33}$ | 'after' |
| | c | bu$^{33}$ | + | dzɯ$^{33}$ | → | bu$^{44}$dzɯ$^{33}$ | 'mate' |
| | d | ɔ$^{33}$ | + | nɔ$^{33}$ | → | ɔ$^{44}$nɔ$^{33}$ | 'if' |
| | e | pu$^{33}$ | + | tʰɯ$^{33}$ | → | pu$^{44}$tʰɯ$^{33}$ | 'Butuo (place name)' |
| | f | ti$^{33}$ 'cloud' | + | nɛ$^{33}$ 'black' | → | tɛ$^{44}$nɛ$^{33}$ | 'dark cloud, nimbostratus' |
| | g | tʂʰɯ$^{33}$ 'rice, oryza sativa' | + | dza$^{33}$ 'grain' | → | tʂʰa$^{44}$dza$^{33}$ | 'rice' |

This sandhi rule can also mark the compounding of the verbs. The rise to tone 44 suggests that it is a compound word and the interpretation is from left to right; see (46). But if the tone is not raised, namely, xɯ$^{33}$dzɯ$^{33}$ and ŋɯ$^{33}$dzɯ$^{33}$, the interpretation of xɯ$^{33}$ and ŋɯ$^{33}$ changes to 'meat' and 'fish', respectively. The expressions are thus understood as phrases, not words, meaning 'to eat the meat' and 'to eat the fish'.

| 46 | a | xɯ$^{33}$ 'to cut off' or 'meat' | + | dzɯ$^{33}$ 'eat' | → | xɯ$^{44}$dzɯ$^{33}$ | 'to cut and eat' |
|----|---|----|---|----|---|----|----|
| | b | ŋɯ$^{33}$ 'to borrow' or 'fish' | + | dzɯ$^{33}$ 'eat' | → | ŋɯ$^{44}$dzɯ$^{33}$ | 'to borrow and eat' |

However, exceptions about this lexical tone sandhi can easily be found in Adur Niesu, such as nɛ$^{33}$su$^{33}$ 'the Niesu people', zɨ$^{33}$lo$^{33}$ 'well, sink', ŋgɯ$^{33}$fu$^{33}$ 'buckwheat pie', and nɛ$^{33}$dʑɨ$^{33}$ 'sun'. This sandhi pattern is also found in Nuosu (see Bradley 1990 for sandhi rules of Nuosu), but with higher productivity than in Adur Niesu. For example, this sandhi rule applies to bo$^{44}$fu$^{33}$ 'cheekbone' in Nuosu, but not to bo$^{33}$fu$^{33}$ 'cheekbone' in Adur Niesu.

This tone sandhi seldom occurs in phrases in Adur Niesu. In (47), where all expressions can be understood as phrases, this sandhi rule does not apply. For example, (47d) does not refer to a particular kind of snake, but a generic term to cover all snakes living or happening to be found in the water. However, this restriction seems less rigid in Nuosu. Gerner (2013) reported that the demonstrative would rise to tone 44 in Nuosu if there was a following classifier of tone 33, such as tsʰɨ$^{44}$ma$^{33}$ (this CLF) 'this one' and tsʰɨ$^{44}$bo$^{33}$ (this CLF.PL) 'these ones'. In contrast, the tone of the demonstrative is not raised in Adur Niesu, namely, tsʰɨ$^{33}$ma$^{33}$ 'this one'. According to the Adur consultants, if the demonstrative is raised in tone, it means emphasis. It is more natural to keep the original tone 33 in this combination.

| 47 | a | zɨ$^{33}$ | 'water, river' | + | dʑi$^{33}$ | 'clean' | → | zɨ$^{33}$dʑi$^{33}$ | 'clean water' |
|----|---|----|----|---|----|----|---|----|----|
| | b | lɯ$^{33}$ | 'cow' | + | tʂʰɨ$^{33}$ | 'manure' | → | lɯ$^{33}$tʂʰɨ$^{33}$ | 'cow's manure' |
| | c | ʁo$^{33}$ | 'bear' | + | tʂɨ$^{33}$ | 'bile' | → | ʁo$^{33}$tʂɨ$^{33}$ | 'bile of the bear' |
| | d | zɨ$^{33}$ | 'water, river' | + | ʂi$^{33}$ | 'snake' | → | zɨ$^{33}$ʂi$^{33}$ | 'snake(s) in the water (not a kind of snake)' |

*4.3. Tone Sandhi: 21 > 44 / 21 _*

This is another relatively productive sandhi rule in Adur Niesu. Similar to the sandhi rule 33 > 44 / _ 33, this rule is again a case of tone dissimilation. Unlike 33 > 44 / _ 33, the sandhi rule 21 > 44 / 21 _ mainly occurs at the phrasal level, such as the auxiliary verb constructions from (48a) to (48c) and the noun phrases from (48d) to (48e). Its effect at the word level is not commonly found in Adur Niesu, for example, si$^{21}$ 'to curse' + tɕʰɯ$^{21}$ 'to revile' → si$^{21}$tɕʰɯ$^{21}$ 'to curse'. If the adjacent tones are different, this sandhi rule does not apply, for example, dzɯ$^{33}$ 'to eat' + do$^{21}$ 'can' → dzɯ$^{33}$do$^{21}$ 'can eat'.

| 48 | a | ndu$^{21}$ | 'hit' | + | to$^{21}$ | 'can' | → | ndu$^{21}$to$^{44}$ | 'can hit' |
|----|---|----|----|---|----|----|---|----|----|
| | b | si$^{21}$ | 'know' | + | to$^{21}$ | 'can' | → | si$^{21}$to$^{44}$ | 'can understand' |
| | c | pu$^{21}$ | 'carry' | + | to$^{21}$ | 'can' | → | pu$^{21}$to$^{44}$ | 'can carry' |
| | d | tsʰɨ$^{21}$ | 'his, her, its' | + | dʑɨ$^{21}$ | 'lower part' | → | tsʰɨ$^{21}$dʑɨ$^{44}$ | 'beneath him/her/it (lit. the part below him/her/it)' |
| | e | ŋa$^{21}$ | 'my' | + | tɕo$^{21}$ | 'direction' | → | ŋa$^{21}$tɕo$^{44}$ | 'to me (lit. my direction)' |

### 4.4. Tone Change in Compounds

Compounding is a productive means of word formation in Adur Niesu. Tone change can serve as a phonological criterion to distinguish compound words from phrases.

#### 4.4.1. Tone 33 > 21/_ zɯ³³

33 > 21 / _ zɯ³³ occurs in compound words of animacy marked with the diminutive marker zɯ³³, grammaticalized from the noun meaning 'son'.

| 49 | a | tʂʰɨ³³ | 'dog' | + | zɯ³³ | → | tʂʰɨ²¹zɯ³³ | 'puppy' |
|----|---|--------|-------|---|------|---|-----------|---------|
|    | b | mu³³ | 'horse' | + | zɯ³³ | → | mu²¹zɯ³³ | 'pony' |
|    | c | ŋɯ³³ | 'fish' | + | zɯ³³ | → | ŋɯ²¹zɯ³³ | 'young fish' |
|    | d | ʂu³³ | 'pheasant' | + | zɯ³³ | → | ʂu²¹zɯ³³ | 'young pheasant' |

If the tone change does not occur, the meaning is also changed, namely, it becomes a nominal–nominal genitive phrase meaning 'the offspring of the animal'.[2] Compare (50) with (49).

| 50 | a | tʂʰɨ³³ | 'dog' | + | zɯ³³ | → | tʂʰɨ³³zɯ³³ | 'dog's offspring' |
|----|---|--------|-------|---|------|---|-----------|---------|
|    | b | mu³³ | 'horse' | + | zɯ³³ | → | mu³³zɯ³³ | 'horse's offspring' |
|    | c | ŋɯ³³ | 'fish' | + | zɯ³³ | → | ŋɯ³³zɯ³³ | 'fish's offspring' |
|    | d | ʂu³³ | 'pheasant' | + | zɯ³³ | → | ʂu³³zɯ³³ | 'pheasant's offspring' |

Moreover, if the compound words with the diminutive marker refer to inanimate beings, such as mountains, the tone change does not occur.

| 51 | a | bo³³ | 'mountain' | + | zɯ³³ | → | bo³³zɯ³³ | 'small mountain' |
|----|---|------|------------|---|------|---|----------|------------------|
|    | b | zɨ³³ | 'water, river' | + | zɯ³³ | → | zɨ³³zɯ³³ | 'small river, creek' |
|    | c | sɨ³³ | 'tree' | + | zɯ³³ | → | sɨ³³zɯ³³ | 'small tree' |
|    | d | tʰa³³ | 'jar' | + | zɯ³³ | → | tʰa³³zɯ³³ | 'small jar' |

This tone change does not happen to all animate beings if there is a ready expression for their offspring. For example, since there is an expression for 'calf', namely, ko³³li³³zɯ³³, lɯ³³zɯ³³ is, therefore, a phrase, meaning 'offspring of the cow', without the tone change. Other examples are:

| | | | | | | | Meaning | Terminology for offspring |
|----|---|------|-----------|---|------|---|---------|--------------------------|
| 52 | a | ʐo³³ | 'sheep' + | zɯ³³ | → | ʐo³³zɯ³³ | 'offspring of the sheep' | ʐo³³la³³zɯ³³ 'lamb' |
|    | b | lɯ³³ | 'cow' + | zɯ³³ | → | lɯ³³zɯ³³ | 'offspring of the cow' | ko³³li³³zɯ³³ 'calf' |
|    | c | ʐɛ³³ | 'chicken' + | zɯ³³ | → | ʐɛ³³zɯ³³ | 'offspring of the chicken' | ʐɛ³³tsɨ⁵⁵zɯ³³ 'chick' |

#### 4.4.2. Tones 33 > 21/_ pa⁵⁵ and 33 > 21/_ pu³³

The two rules of tone change are discussed together since both of them occur in similar semantic environment, namely, about the masculine gender of animate beings. The words are compounded with an animal formative and the masculine morpheme pa⁵⁵ and pu³³. Morpheme pa⁵⁵ is a reflex of PTB *p/ba 'male, father, 3ʳᵈ pronoun' and pu³³ is of PTB *pu 'male, masculine suffix' (see Matisoff 2003). Adur Niesu uses the former to refer to 'parents', namely, pʰa⁵⁵mo⁵⁵, with additional aspiration. Both pa⁵⁵ and pu³³ will cause the preceding 33 tone to be lowered. The dog word tʂʰɨ³³ can go with either masculine morpheme, and its tone is lowered in both compounding; see (53a) and (53g). Bearing the male morpheme pa⁵⁵, the word 'horse' mu²¹pa⁵⁵ has extended its meaning to cover both male and female horses. As a consequence, another gender morpheme is needed to specify whether it is a male or female horse in modern Adur Niesu, namely, mu²¹pu³³ 'male horse' and mu²¹mo²¹ 'female horse'. In some cases, the masculine marker pu³³ is voiced, such as in lɛ²¹bu³³ 'ox', but the tone change rule still holds.

However, if the preceding syllable bears the 55 tone, it will not be lowered due to the masculine syllable, for example, tʂʰɨ⁵⁵bu³³ 'male goat' and vi⁵⁵pa⁵⁵ 'female pig'.

| 53 | | 33 > 21/ _ pa$^{55}$ | | | | | | |
|---|---|---|---|---|---|---|---|---|
| | a | tʂʰɨ$^{33}$ | 'dog' | + | pa$^{55}$ | → | tʂʰɨ$^{21}$pa$^{55}$ | 'male dog' |
| | b | mu$^{33}$ | 'horse' | + | pa$^{55}$ | → | mu$^{21}$pa$^{55}$ | 'male horse, horse' |
| | | 33 > 21/ _ pu$^{33}$ | | | | | | |
| | c | fɛ$^{33}$ | 'mouse' | + | pu$^{33}$ | → | fɛ$^{21}$pu$^{33}$ | 'male mouse' |
| | d | mu$^{33}$ | 'horse' | + | pu$^{33}$ | → | mu$^{21}$pu$^{33}$ | 'male horse' |
| | e | ʂu$^{33}$ | 'pheasant' | + | pu$^{33}$ | → | ʂu$^{21}$pu$^{33}$ | 'male pheasant' |
| | f | fa$^{33}$ | 'golden pheasant' | + | pu$^{33}$ | → | fa$^{21}$pu$^{33}$ | 'male golden pheasant' |
| | g | tʂʰɨ$^{33}$ | 'dog' | + | pu$^{33}$ | → | tʂhi$^{21}$pu$^{33}$(tʂʰɨ$^{21}$mo$^{21}$) | 'male dog (and female dog)' |

### 4.4.3. Tone 33 > 21/_ mo$^{21}$

This tone change occurs if the preceding syllable bearing tone 33 is followed by the feminine morpheme mo$^{21}$, a reflex of Proto-Loloish *ʔəC-ma³ 'mother' (Bradley 1979). Like many Tibeto–Burman languages, Adur Niesu mo$^{21}$ can also function as an augmentative morpheme (see Matisoff 1992). This rule of tone change is effective if mo$^{21}$ is used for two functions, i.e., a feminine marker and an augmentative marker, regardless of the animacy of the word. If the preceding syllable does not bear tone 33, this tone change does not apply, such as vi$^{55}$mo$^{21}$ (pig:female) 'female pig' and tɕi$^{55}$mo$^{21}$ (eagle:female) 'female eagle'.

| 54 | a | mu$^{33}$ | 'horse' | + | mo$^{21}$ | → | mu$^{21}$mo$^{21}$ 'female horse' |
|---|---|---|---|---|---|---|---|
| | b | zo$^{33}$ | 'sheep' | + | mo$^{21}$ | → | zo$^{21}$mo$^{21}$ 'female sheep' |
| | c | luɯ$^{33}$ | 'cow' | + | mo$^{21}$ | → | luɯ$^{21}$mo$^{21}$ 'female cow' |
| | d | ʁo$^{33}$ | 'bear' | + | mo$^{21}$ | → | ʁo$^{21}$mo$^{21}$ 'female bear' |
| | e | tʰa$^{33}$ | 'jar' | + | mo$^{21}$ | → | tʰa$^{21}$mo$^{21}$ 'big jar' |
| | f | zɨ$^{33}$ | 'water' | + | mo$^{21}$ | → | zɨ$^{21}$mo$^{21}$ 'big river' |
| | g | bo$^{33}$ | 'mountain' | + | mo$^{21}$ | → | bo$^{21}$mo$^{21}$ 'big mountain' |
| | h | pi$^{33}$ | 'priest' | + | mo$^{21}$ | → | pi$^{21}$mo$^{21}$ 'big (highly experienced) priest' |

Similar to the masculine marker pa$^{55}$ in mu$^{21}$pa$^{55}$ which covers both male and female horse as a general term, mo$^{21}$ can also be lexicalized with its feminine meaning being implicit, such as dʑɯ$^{21}$mo$^{21}$ 'bee, queen bee'. But this rule of tone change still holds because of the feminine marker.

However, this tone change does not apply to other meanings derived from mo$^{21}$. In Adur Niesu, besides 'female' and 'big', mo$^{21}$ can also function as a nominal meaning 'woman' and 'master', such as nɛ$^{33}$mo$^{21}$ (black Yi:woman) 'the women of the Black Yi (the historical noble class)', ma$^{55}$mo$^{21}$ (teach:master) 'teacher', and a postposed modifier meaning 'old', such as tsʰo$^{33}$mo$^{44}$ (people:old) 'old people' and ʁo$^{33}$mo$^{44}$ (bear:old) '(old) bear'. This tone change does not apply to the above three meanings. Note the contrast between pi$^{33}$mo$^{44}$ 'priest' and pi$^{21}$mo$^{21}$ 'big (highly experienced) priest'. The former is a general term and also the title to refer to a Yi priest, and the latter is only used for priests with experiences and achievements. For example, while dʑɯ$^{33}$kʰɯ$^{33}$pi$^{33}$mo$^{44}$ means simply 'Priest Jike', pi$^{21}$mo$^{21}$dʑɯ$^{33}$kʰɯ$^{33}$ is a nominal–nominal phrase, meaning 'Jike, the highly experienced and accomplished priest'. Additionally, this rule of tone change serves as a criterion to distinguish two confusing meanings in Adur Niesu, namely, 'old' and 'big'. In many languages of the world, 'old' and 'big' can be colexified (Rzymski et al. 2020). If this tone change occurs in compound words, the meaning is not 'old', but 'big', for example, sɨ$^{21}$mo$^{21}$ 'big tree'. To express 'old tree', a phrase is needed, namely, sɨ$^{33}$ a$^{33}$mo$^{21}$ (tree old) 'old tree'.

### 4.4.4. Tone 33 > 21/_ ɲi$^{55}$

This rule of tone change occurs in the semantic environment of dual marking, with the plural pronouns compounded with the dual morpheme ɲi$^{55}$.

| 55 | a | tʰu$^{33}$ | 'they' | + | ɲi$^{55}$ | → | tʰu$^{21}$ɲi$^{55}$ | 'the two of them' |
|---|---|---|---|---|---|---|---|---|
| | b | no$^{33}$ | 'you (PL)' | + | ɲi$^{55}$ | → | no$^{21}$ɲi$^{55}$ | 'the two of you' |
| | c | ŋo$^{33}$ | 'we (exclusive)' | + | ɲi$^{55}$ | → | ŋo$^{21}$ɲi$^{55}$ | 'the two of us (exclusive)' |

It should be noted that the dual marker ȵi$^{55}$ is derived from, but different from, the cardinal word ȵi$^{21}$ 'two'. This can be proved by the evidence from a$^{33}$si$^{55}$ȵi$^{55}$ (1PL.inclusive dual) 'the two of us (inclusive)' where, without tone 33 on the preceding syllable, the dual marker still bears tone 55, not tone 21. Otherwise, ȵi$^{21}$ will be considered to colexify 'dual' and 'two', which is an unlikely proposal for Adur Niesu.

### 4.4.5. Tone 33 > 44/ ha$^{21}$ _

This tone change occurs in interrogatives of quantity, such as 'how many' and 'how long'. The interrogative words are compounds, formed by the interrogative morpheme ha$^{21}$ and the adjectival roots; see Table 16. Both ha$^{21}$ and the adjectival roots are bound morphemes, and cannot be used as full words. This tone change is also found in Nuosu; see Table 16.

**Table 16.** Adur Niesu and Nuosu interrogatives of quantity.

| Meaning | Shynra Nuosu | Adur Niesu |
|---|---|---|
| how big? | kʰɯ$^{21}$ʐɿ$^{44}$ | ha$^{21}$ʐɿ$^{44}$ |
| how thick (e.g., tree, string)? | kʰɯ$^{21}$fu$^{44}$ | ha$^{21}$fu$^{44}$ |
| how high? | kʰɯ$^{21}$m̥u$^{44}$ | ha$^{21}$mu$^{44}$ |
| how long (distance)? | kʰɯ$^{21}$ʂo$^{44}$ | ha$^{21}$ʂɯ$^{44}$ |
| how long (time)? | kʰɯ$^{21}$ho$^{44}$ | ha$^{21}$ȵo$^{44}$ |
| how wide (2-dimensional)? | kʰɯ$^{21}$fi$^{44}$ | ha$^{21}$fi$^{44}$ |
| how wide (3-dimensional)? | kʰɯ$^{21}$dʑi$^{44}$ | ha$^{21}$dʐɿ$^{44}$ |
| how thick? | kʰɯ$^{21}$tu$^{44}$ | ha$^{21}$tʰu$^{44}$ |
| how many? | kʰɯ$^{21}$ȵi$^{44}$ | ha$^{21}$ȵo$^{44}$ |
| how heavy? | kʰɯ$^{21}$li$^{44}$ | ha$^{21}$li$^{44}$ |

Adur Niesu ha$^{21}$ should follow the derivational chain from the category of selection a$^{21}$si$^{21}$ 'which' to the category of manner ha$^{21}$mu$^{33}$ (how:do) 'how'.

First, typologically, the derivational direction from the interrogative category of selection to that of manner is attested, not the other way around (see Hölzl 2018, p. 83). Second, Adur Niesu ha$^{21}$ 'how' and a$^{21}$si$^{21}$ 'which' are closely related; the former should be a form after syllable reduction of the latter. After the syllable reduction of si from a$^{21}$si$^{21}$, a fricative glottal /h/ can often be epenthesized, such as ha$^{21}$ʐɿ$^{44}$ / a$^{21}$ʐɿ$^{44}$ 'how big?' and ha$^{21}$ȵo$^{44}$ / a$^{21}$ȵo$^{44}$ 'how many?'. The epenthesized form now becomes the dominant form of this morpheme. A similar epenthesis is shown in (36).

Adur Niesu ha$^{21}$ can be interchangeably pronounced as a$^{21}$si$^{21}$ as in a$^{21}$si$^{21}$mu$^{33}$ / ha$^{21}$mu$^{33}$ (how:do) 'how' and as a$^{21}$si$^{21}$tʰɯ$^{33}$ / ha$^{21}$tʰɯ$^{33}$ (which:time) 'when'. Therefore, a$^{21}$si$^{21}$ means both 'which' and 'how' in Adur Niesu. Treating a$^{21}$si$^{21}$ as the *how* form in Adur Niesu is attested by PL *ʔəs (Bradley 1979, p. 334). The Nuosu kʰɯ$^{21}$ should be a reflex of the Proto-TB *ka (Matisoff 2003). Unlike Adur Niesu, Nuosu kʰɯ$^{21}$ has lost its etymological connection with its modern *which* word ɕi$^{44}$ (Shynra) and ɕa$^{42}$ (Yynuo). The possible reason is that, at a certain historical moment, there used to be two *which* words in Nuosu: the canonical *which* lexeme, cognate of the Proto-TB *ka, and an innovation derived from other interrogatives (e.g., *where* and *what*). Gradually, the innovative form replaced the old *which* lexeme (Ding 2022).

Functioning as the interrogative category of selection, or 'which', a$^{21}$si$^{21}$ is an adjective, placed after the head noun, such as tsʰo$^{33}$ a$^{21}$si$^{21}$ ma$^{33}$ (people, which CLF) 'which person'. Due to its being used for another function, namely, the interrogative of manner, the *which* word a$^{21}$si$^{21}$ has changed its adjectival word class, and is used as an adverb in the *how* word, placed before verbs, namely, a$^{21}$si$^{21}$mu$^{33}$ (how:do) or ha$^{21}$mu$^{33}$ (how:do) 'how'. As a consequence, after the functional change, it is no longer acceptable to pronounce ha$^{21}$ȵo$^{44}$ 'how many/much' as *a$^{21}$si$^{21}$ȵo$^{44}$ or a$^{21}$si$^{21}$ ma$^{33}$ 'which one' as *ha$^{21}$ ma$^{33}$ in Adur Niesu. The irreversibility between ha$^{21}$ and a$^{21}$si$^{21}$ in the selection interrogative and the quantity interrogative suggests that ha$^{21}$ has become a different morpheme

with different word class and different function from a$^{21}$si$^{21}$ 'which', although it is derived from the *which* morpheme. The derivational path in Adur Niesu is proposed as below (see Figure 7 and also Ding 2022).

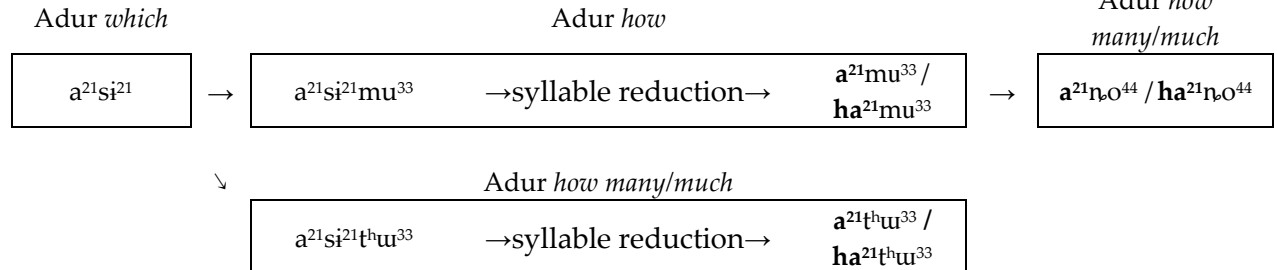

**Figure 7.** The derivation of Adur Niesu *which*, *how*, and related categories.

### 4.5. Tone Change in Prefixed Words

Tone change occurs to the prefixes a$^{33}$- and i$^{33}$-, which are used in the formation of property-denoting words, kinship terms, and animal words.

### 4.5.1. Tone 33 > 44/ _ 33 in Dimensional Words

This tone change is most popular in a$^{33}$-/i$^{33}$- prefixed property-denoting words in Adur Niesu.

The words in Table 17 are called stative verbs of dimensional extentives in Bradley (1995). In modern Adur Niesu, the positive dimensional words are prefixed by a$^{33}$-, and the negative ones are prefixed by i$^{33}$-, both sharing the same root. This derivational pattern is not productive in modern Nuosu and Niesu. However, historically, the positive and negative forms may have different roots, such as Nuosu a$^{44}$li$^{33}$ 'heavy' and ʐo$^{44}$so$^{33}$ 'light', and Adur Niesu a$^{44}$ʐɿ$^{33}$ 'big' and ɛ$^{55}$tsi$^{33}$ 'small'. According to Bradley (1995), the historical development is that the original negative dimensional words were replaced by forms that have the prefix i$^{33}$- plus the positive dimensional words. The negative dimensional word 'small' in the big/small pair has persisted and has not been replaced by the i$^{33}$-prefixed positive form in Nuosu and Niesu in Table 17. In the cases of 'heavy/light', while the replacement of the negative extentive forms by the positive forms occurs in Adur Niesu, the negative forms ʐo$^{44}$so$^{33}$ (Shynra Nuosu), or i$^{33}$so$^{33}$ (Yynuo Nuosu) 'light', have survived. But a different prefix ʐo$^{33}$-, rather than i$^{33}$-, is added to so$^{33}$ 'light' in Shynra Nuosu (see Ding 2022).

**Table 17.** Adur Niesu dimensional words.

| Meaning | Shynra Nuosu | Adur Niesu |
|---|---|---|
| big | a$^{44}$ʐɿ$^{33}$ | a$^{44}$ʐɿ$^{33}$ |
| small | ɛ$^{55}$tsi$^{33}$ | ɛ$^{55}$tsi$^{33}$ |
| thick (e.g., tree, string) | a$^{33}$fu$^{33}$ | a$^{44}$fu$^{33}$ |
| slender | i$^{44}$fu$^{33}$ | i$^{44}$fu$^{33}$ |
| high | a$^{33}$m̩u$^{33}$ | a$^{44}$mu$^{33}$ |
| low | i$^{44}$m̩u$^{33}$ | i$^{44}$mu$^{33}$ |
| long (distance) | a$^{33}$ʂo$^{33}$ | a$^{44}$ʂɯ$^{33}$ |
| short | i$^{44}$ʂo$^{33}$ | i$^{44}$ʂɯ$^{33}$ |
| long (time) | a$^{33}$ho$^{44}$ | a$^{44}$ȵo$^{33}$ |
| short | i$^{33}$ho$^{44}$ | i$^{44}$ȵo$^{33}$ |
| wide (2-dimensional) | a$^{33}$fi$^{33}$ | a$^{44}$fi$^{33}$ |
| narrow | i$^{44}$fi$^{33}$ | i$^{44}$fi$^{33}$ |

**Table 17.** *Cont.*

| Meaning | Shynra Nuosu | Adur Niesu |
|---|---|---|
| wide (3-dimensional) | $a^{33}dʑi^{33}$ | $a^{44}dʑi^{33}$ |
| narrow | $i^{44}dʑi^{33}$ | $i^{44}dʑi^{33}$ |
| thick (e.g., book) | $a^{33}tu^{33}$ | $a^{44}t^hu^{33}$ |
| thin | $i^{44}tu^{33}$ | $i^{44}t^hu^{33}$ |
| many | $a^{44}ɲi^{33}$ | $a^{44}ɲo^{33}$ |
| few | $i^{44}ɲi^{33}$ | $i^{44}ɲo^{33}$ |
| heavy | $a^{44}li^{33}$ | $a^{44}li^{33}$ |
| light | $ʐo^{44}so^{33}$ | $i^{44}li^{33}$ |

It can be observed that this tone change spreads to all dimensional extentives in Adur Niesu, but not in Nuosu.

4.5.2. Tone 33 > 44/ _ 33 in Kinship and Animal Words

This tone change is also related to the prefix $a^{33}$- in other word formations besides the dimensional words. Although many of them have lost productivity, historically, it has several other semantic functions in Adur Niesu, including kinship terms, color words, and animal words. See Matisoff (2018) for a cross-linguistic study of Proto-Tibet—Burman *a-prefix.

In modern Adur Niesu, this tone change only has certain productivity in kinship terms and animal names, besides the dimensional words. Given names of Adur Niesu are mostly bisyllabic, such as $ga^{33}ko^{33}$, a given name often for female. One of the syllables of the given name can be taken and prefixed by $a^{33}$- to express endearment with this tone change, such as $a^{44}ko^{33}$.

56  a  a-prefixed kinship terms
$a^{44}bo^{33}$                    'father's sister'
$a^{44}ta^{33}$                    'father'
$a^{44}p^hu^{33}$                   'grandfather'
$a^{44}mo^{33}$                    'mother'

  b  a-prefixed endearment addresses
$a^{44}ko^{33}$              often for female
$a^{44}si^{33}$              often for female
$a^{44}ga^{33}$              often for male
$a^{44}t^hi^{33}$             often for male
$a^{44}ndza^{33}$            both for female
                and male

This rule of tone change still applies to a large number of animal words, with some exceptions (e.g., $a^{33}ʁo^{44}$ 'bear' and $a^{21}dʑa^{33}$ 'sparrow').

57  a  $a^{44}ɲɛ^{33}$                         'cat'
  b  $a^{44}fɛ^{33}$                         'mouse'
  c  $a^{44}lɛ^{33}$                         'goat'
  d  $a^{44}du^{33}$                         'fox'
  e  $a^{44}ɬi^{33}$                         'pigeon'
  f  $a^{44}dʐɯ^{33}$                        'raven'
  g  $a^{44}ʁo^{33}$                         'hoopoe bird'

The a-prefix in color terms are lexicalized without any tone change, such as $a^{33}ni^{33}$ 'red, be red', $a^{33}t^hu^{33}$ 'white, be white', $a^{33}nɛ^{33}$ 'black, be black', and $a^{33}ʂi^{33}$ 'yellow, be yellow'. If the tone change rules apply, the meanings will be changed. For example, the consultants indicate that $a^{44}t^hu^{33}$, with the tone of the prefix raised to 44, means 'thick (e.g., book)' (see Table 17), but not 'white, be white' anymore.

### 4.6. Tone Change in Patient Marking

There are three rules of tone change about patient marking, which are discussed together: patient$^{33}$ > 44 / _ 33; patient$^{33}$ > 21 / _ ko$^{33}$; 21 > 44 / patient$^{33}$_.

Since Adur Niesu is SOV, if there is only one argument in the clause, it could be agent or patient. In some cases, the default context is clear to tell the meaning, such as xɯ$^{33}$dzɯ$^{33}$ (meat eat) 'to eat the meat' as a non-reversible event. However, in many cases, ambiguity emerges. To disambiguate, other than the contexts, there are two main means to mark the patient of the clause.

First, the tone change of patient$^{33}$ > 44 / _ 33 is addressed. This tone change is on the patient. The argument is mostly monosyllabic personal pronouns before the main verb. The patient will change from tone 33 to tone 44. This strategy is often used when the main verb bears tone 33.

| 58 | nɯ$^{33}$ | hi$^{21}$ | **ŋo$^{44}$** | kɯ$^{33}$. |
|---|---|---|---|---|
| | 2SG | say | 1PL | make listen |
| | 'You tell us (of it).' | | | |

| 59 | i$^{33}$ | | **ŋa$^{44}$** | ɣɯ$^{33}$=a$^{21}$=da$^{33}$. |
|---|---|---|---|---|
| | SG.LOG | | 1SG | win=NEG=SP |
| | 'He/she cannot win over me.' | | | |

Compare (60a) and (60b). With the tone change, the ambiguity in (60a) can be eliminated in (60b). Despite the ambiguity in (60a), it will often be understood, without the tone change, as a resultative construction 'someone stole his (belongings)' in Adur Niesu, where the patient is placed sentence initially as the topic and the rest the comment.[3]

Due to the analytic morphology of Adur Niesu, there is the possibility that this tone change is caused by some floating tone marking patient. However, since we do not have any supporting evidence, it is synchronically an issue of tone change.

| 60 | a | tsʰi̱$^{33}$ | tsʰo$^{33}$ | kʰu$^{33}$. |
|---|---|---|---|---|
| | | 3SG | people | steal |
| | | 'Someone stole his (belongings).' or 'he stole someone's (belongings).' | | |
| | b | tsʰi̱$^{33}$ | tsʰo$^{44}$ | kʰu$^{33}$. |
| | | 3SG | people.P | steal |
| | | 'He stole someone's (belongings).' | | |
| | c | tsʰi̱$^{33}$ | tsʰo$^{21}$=ko$^{33}$ | kʰu$^{33}$. |
| | | 3SG | people=DOM | steal |
| | | 'He stole someone's (belongings).' | | |

If the monosyllabic arguments are replaced by polysyllabic ones, the tone change cannot apply. To disambiguate (61a), an analytic means by differential object marker ko$^{33}$ is used to mark the patient; see (61b). Similarly, since there is the way to disambiguate, (61a), without the marking, it will often be understood as 'dʑɛ$^{21}$nɛ$^{33}$ stole su̱$^{33}$ga$^{55}$'s (belongings)'.

| 61 | a | su̱$^{33}$ga$^{55}$ | dʑɛ$^{21}$nɛ$^{33}$ | kʰu$^{33}$. |
|---|---|---|---|---|
| | | surname | surname | steal |
| | | 'dʑɛ$^{21}$nɛ$^{33}$ stole su̱$^{33}$ga$^{55}$'s (belongings).' or 'su̱$^{33}$ga$^{55}$ stole dʑɛ$^{21}$nɛ$^{33}$'s (belongings).' | | |
| | b | su̱$^{33}$ga$^{55}$ | dʑɛ$^{21}$nɛ$^{33}$=ko$^{33}$ | kʰu$^{33}$. |
| | | surname | surname=DOM | steal |
| | | 'su̱$^{33}$ga$^{55}$ stole dʑɛ$^{21}$nɛ$^{33}$'s (belongings).' | | |

The differential object marker (DOM) can also be used with monosyllabic patients for disambiguation; see (60c). In this case, the tone change rule of patient$^{33}$ > 21 / _ ko$^{33}$ is applied. The citation tone 33 of the person pronoun will be lowered to tone 21; see (62). The tone lowering or dissimilation before the DOM occurs regardless of the tonal value of the main verb.

62  a  tsʰɿ³³                          tsʰo³³                          si⁵⁵.

       3SG                            people                         kill

       'Someone killed him.' or 'He killed someone'

    b  tsʰɿ³³                          tsʰo²¹=ko³³                     si⁵⁵.

       3SG                            people=DOM                     kill

       'He killed someone.'

    c  tsʰɿ³³                          tsʰo³³                          vi⁵⁵.

       3SG                            people                         carry on shoulder

       'Someone shouldered him.'  or 'He shouldered someone'

    d  tsʰɿ³³                          tsʰo²¹=ko³³                     vi⁵⁵.

       3SG                            people=DOM                     carry on shoulder

       'He shouldered someone.'

    e  xo³³tʰi⁵⁵ɬa²¹ba³³              tsʰɿ²¹=ko³³                     a²¹=hi²¹.

       name                           3SG=DOM                        NEG=say

       'Hotihlabba ignored him.'

    f  nɯ³³                            ŋa²¹=ko³³            pʰu⁴⁴            la³³.

       2SG                            1SG=DOM             save            come

       'You come to save me.'

While the above two rules of tone change apply to the argument, the tone change 21 > 44 / patient³³_ applies to the main verb; see Table 18.

**Table 18.** Adur Niesu argument marking through tone.

|                          | Meaning        | Agent marking | Patient marking |
| ------------------------ | -------------- | ------------- | --------------- |
| argument + ndu²¹         | 'to beat'      | ndu²¹         | ndu⁴⁴           |
| argument + ʂɯ²¹          | 'to find'      | ʂɯ²¹          | ʂɯ⁴⁴            |
| argument + pu²¹          | 'to carry'     | pu²¹          | pu⁴⁴            |
| argument + bi²¹          | 'to give'      | bi²¹          | bi⁴⁴            |
| argument +nɯ²¹           | 'to chase'     | nɯ²¹          | nɯ⁴⁴            |

Specifically, if the main verb bears tone 21, to mark the patient, the original tone 21 of the main verb will be raised to tone 44, suggesting the preceding argument is the patient, no longer the agent. In (63a), the citation form 'to find, search' in Adur Niesu is ʂɯ²¹. If it is changed to tone 44, the preceding pronoun becomes the patient; see (63b). It is also acceptable to use the DOM in (63c) with the patient changing its tone to 21. Please note that tonal rising for patient marking does not occur to the main verb bearing tone 33, such as ku³³ 'to steal', and such a sentence is not acceptable, i.e., *tsʰɿ³³ tsʰo³³ kʰu⁴⁴ (3SG people steal, intended meaning: 'he stole someone's belongings').

63  a  ŋa³³                     ʂɯ²¹                     o⁴⁴.

       1SG                      find                     PFV

       'I searched (, but in vain).'

    b  tsʰɿ³³                    ŋa³³                     ʂɯ⁴⁴.

       3SG                      1SG                      find

       'He (is) looking for me.'

    c  tsʰɿ³³                    ŋa²¹=ko³³                ʂɯ⁴⁴.

       3SG                      1SG=DOM                  find

       'He (is) looking for me.'

The following pairs are only contrastive in the tone of the verb. If the original tone 21 is changed to 44, the meaning is also changed; see (64) to (66).

64  a  tsʰo³³                          **ndu²¹**=ʂi³³                     ŋu³³.

       people                         beat=NMLZ                         COP

       '(This wound) is (caused) by (someone's) beating.'

    b  tsʰo³³                          **ndu⁴⁴**=ʂi³³                     ŋu³³.

       people                         beat=NMLZ                         COP

       '(This is) something (used) to beat people'

65　a　a$^{44}$ta$^{33}$　　**pu$^{21}$**=nɯ$^{44}$=ɕi$^{44}$　zɨ$^{33}$　　　　　kɯ$^{33}$　　　la$^{33}$.

　　　　father　　　　carry=IMPF=SEQ　　water　　　　　throw　　　come

　　　　'Father carried (something) and threw into the water.'

　　b　a$^{44}$ta$^{33}$　　**pu$^{44}$**=nɯ$^{44}$=ɕi$^{44}$　zɨ$^{33}$　　　　　kɯ$^{33}$　　　la$^{33}$.

　　　　father　　　　carry=IMPF=SEQ　　water　　　　　throw　　　come

　　　　'(Someone) carried the father and threw (him) into the water.'

66　a　ŋa$^{33}$　　　　　　**bɨ$^{21}$**　　　　　　o$^{44}$.

　　　　1SG　　　　　　　give　　　　　　　PFV

　　　　'I gave (it to someone).'

　　b　ŋa$^{33}$　　　　　　**bɨ$^{44}$**　　　　　　o$^{44}$.

　　　　1SG　　　　　　　give　　　　　　　PFV

　　　　'Something (was) given to me.'

### 4.7. Tone Change in Reduplication for Interrogation

There are two rules for tone change to generate reduplication for yes–no interrogations: 33 > 44 / _ 33 and 21 > 33 / 21 _. It is clear that the two rules are consistently tone dissimilation, namely, adjacent same tones trigger dissimilation.

The first tone change, namely, 33 > 44 / _ 33, is productive in reduplicating monosyllabic verbs for yes–no questions; see (67). The first monosyllabic verb with tone 33 will rise to tone 44. See Figure 8 for the tone change.

67　a　zɨ$^{33}$　　　+　　　zɨ$^{33}$　　　→　　　zɨ$^{44}$~zɨ$^{33}$　　　　'to buy or not'

　　b　ndo$^{33}$　　+　　　ndo$^{33}$　　→　　　ndo$^{44}$~ndo$^{33}$　　　'to drink or not'

　　c　la$^{33}$　　　+　　　la$^{33}$　　　→　　　la$^{44}$~la$^{33}$　　　　'to come or not'

　　d　tɕo$^{33}$　　+　　　tɕo$^{33}$　　→　　　tɕo$^{44}$~tɕo$^{33}$　　　'to turn or not'

The tone change rule is not applicable to disyllabic or multisyllabic verbs for interrogative. Therefore, it serves as a criterion to distinguish words and phrases in Adur Niesu. While (68a) and (68b) are verbs without the tone change, (68c) and (68d) are verb phrases with the tone change.

68　a　ɬɛ$^{33}$pʰɔ$^{33}$　'to fight back'　　+　pʰɔ$^{33}$　→　ɬɛ$^{33}$pʰɔ$^{33}$~pʰɔ$^{33}$　　'to fight back or not'

　　b　hi$^{33}$tɕʰi$^{33}$　'to fall down'　　+　tɕʰi$^{33}$　→　hi$^{33}$tɕʰi$^{33}$~tɕʰi$^{33}$　　'to fall down or not'

　　c　zɨ$^{33}$ndo$^{33}$　'to drink water'　+　ndo$^{33}$　→　zɨ$^{33}$ndo$^{44}$~ndo$^{33}$　　'to drink water or not'

　　d　dzɯ$^{33}$tɕʰi$^{33}$　'want eat (something)'　+　tɕʰi$^{33}$　→　dzɯ$^{33}$tɕʰi$^{44}$~tɕʰi$^{33}$　　'to want or not want to eat'

Another rule of tone change found in interrogations, namely, 21 > 33 / 21 _, differs from 33 > 44 / _ 33 in that it occurs in both word and phrase. For example, (69h) and (69i) are words and (69j) is a phrase; the tone change 21 > 33 / 21 _ is still applicable.

As was discussed in Section 3.6, on the surface, there seems to be a third rule of tone change regarding reduplication for interrogation: 55 > 21 / 55 _. However, the tone lowering from 55 to 21 is not a tone change, but the result of the floating tone associated with the interrogative particle a$^{21}$ after syllable reduction.

69　a　hi$^{21}$ 'to say'　　　　　　+　hi$^{21}$　→　hi$^{21}$~ hi$^{33}$　　　　　'to say or not'

　　b　ʂɯ$^{21}$ 'to find'　　　　　+　ʂɯ$^{21}$　→　ʂɯ$^{21}$~ʂɯ$^{33}$　　　'to find or not'

　　c　vu$^{21}$ 'to sell'　　　　　+　vu$^{21}$　→　vu$^{21}$~vu$^{33}$　　　　'to sell or not'

　　d　gɯ$^{21}$ 'to play'　　　　　+　gɯ$^{21}$　→　gɯ$^{21}$~gɯ$^{33}$　　　'to play or not'

　　e　su$^{21}$ 'to resemble'　　　+　su$^{21}$　→　su$^{21}$~su$^{33}$　　　　'to resemble or not'

　　f　ndu$^{21}$ 'to hit'　　　　　+　ndu$^{21}$　→　ndu$^{21}$~ndu$^{33}$　　'to hit or not'

　　g　ŋo$^{21}$ 'to think'　　　　　+　ŋo$^{21}$　→　ŋo$^{21}$~ŋo$^{33}$　　　　'to think or not'

　　h　a$^{33}$go$^{21}$ 'empty'　　　　+　go$^{21}$　→　a$^{33}$go$^{21}$~go$^{33}$　　　'be empty or not'

　　i　mo$^{33}$ŋgo$^{21}$ 'to undo the curse'　+　ŋgo$^{21}$　→　mo$^{33}$ŋgo$^{21}$~ŋgo$^{33}$　'to undo the curse or not'

　　j　xɯ$^{33}$vu$^{21}$ 'to sell the meat'　+　vu$^{21}$　→　xɯ$^{33}$vu$^{21}$~vu$^{33}$　'to sell the meat or not'

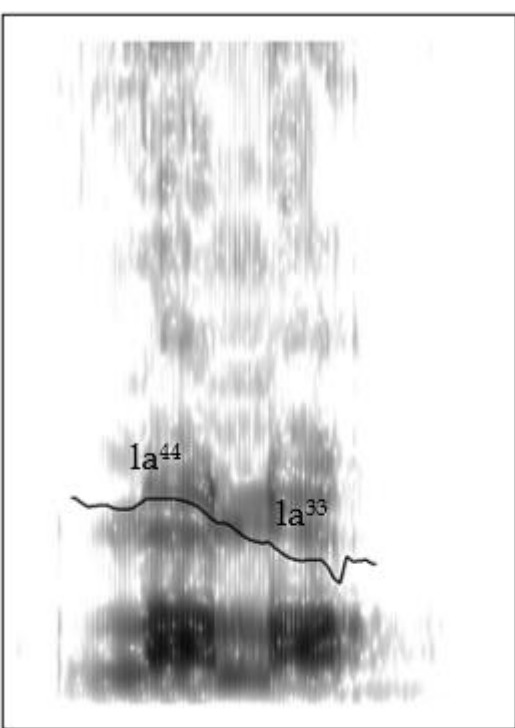

**Figure 8.** The tone change with monosyllabic verb la$^{33}$ 'come'.

*4.8. Effect of Floating Tone*

Finally, the effect of the floating tone is discussed. On the surface, it appears to be a kind of tonal alternation. However, different from tone sandhi and tone change, it is the effect of the tone of an additional syllable after syllable reduction, such as tone 21 left after the reduced interrogative particle a$^{21}$ in Section 3.6.

Another case of floating tone is about tone 21 in Adur Niesu possessive pronouns. Tone 21 was originally borne by the Proto-Nuosu proper genitive marker *ni$^{21}$. This genitive marker is reduced in Adur Niesu and Nuosu, but still kept in Yynuo Nuosu as ni$^{42}$, such as a$^{33}$p$^{h}$u$^{33}$=ni$^{42}$ t$^{h}$ɯ$^{42}$zɿ$^{33}$ (grandfather=ɢᴇɴ book) 'grandfather's book'. Lama (2022) reports the tonal change from Proto-Nuosu proper 21 to modern Yynuo Nuosu 42.

Therefore, the genitive marker overrides its floating tone to Adur Niesu plain personal pronouns, e.g., ŋa$^{33}$ + *ni$^{21}$ → ŋa$^{21}$ 'my'. Take the noun phrases of locational description for example, modified by the possessive pronouns. Adur Niesu locational concepts are mainly expressed through nouns, such as ɖʐɿ$^{21}$ 'lower part' and ŋi$^{33}$ 'front'. Most examples in (70) also experience tone dissimilation, namely, 21 > 44 / 21 _ in Section 4.3.

| | | | | | | |
|---|---|---|---|---|---|---|
| 70 a | ts$^{h}$ɿ$^{21}$ 'his, her, its' | + | ɖʐɿ$^{21}$ 'lower part' | → | ts$^{h}$ɿ$^{21}$ɖʐɿ$^{44}$ | 'beneath him/her/it (lit. the part below him/her/it)' |
| | ts$^{h}$ɿ$^{21}$ 'his, her, its' | + | tɕo$^{21}$ 'direction' | → | ts$^{h}$ɿ$^{21}$tɕo$^{44}$ | 'to him/her/it (lit. his/her/its direction)' |
| | ts$^{h}$ɿ$^{21}$ 'his, her, its' | + | ŋi$^{33}$ 'front' | → | ts$^{h}$ɿ$^{21}$ŋi$^{33}$ | 'in front of him/her/it (lit. his/her/its front)' |
| b | ŋa$^{21}$ 'my' | + | ɖʐɿ$^{21}$ 'lower part' | → | ŋa$^{21}$ɖʐɿ$^{44}$ | 'beneath me (lit. the part below me)' |
| | ŋa$^{21}$ 'my' | + | tɕo$^{21}$ 'direction' | → | ŋa$^{21}$tɕo$^{44}$ | 'to me (lit. my direction)' |
| | ŋa$^{21}$ 'my' | + | ŋi$^{33}$ 'front' | → | ŋa$^{21}$ŋi$^{33}$ | 'in front of me (lit. my front)' |
| c | nɯ$^{21}$ 'your (sing.)' | + | ɖʐɿ$^{21}$ 'lower part' | → | nɯ$^{21}$ɖʐɿ$^{44}$ | 'beneath you (lit. the part below you)' |
| | nɯ$^{21}$ 'your (sing.)' | + | tɕo$^{21}$ 'direction' | → | nɯ$^{21}$tɕo$^{44}$ | 'to you (lit. your direction)' |
| | nɯ$^{21}$ 'your (sing.)' | + | ŋi$^{33}$ 'front' | → | nɯ$^{21}$ŋi$^{33}$ | 'in front of you (lit. your front)' |

## 5. Conclusions

This study describes the segmental and suprasegmental phonology of Adur Niesu, a Loloish (or Ngwi) language spoken mainly in Liangshan, Sichuan, in southwest China. There are 41 phonemic consonants: nine plain plosives, three prenasalized plosives, eleven fricatives, four nasals, two laterals, nine affricates and three prenasalized affricates. Com-

pared with Nuosu, a close dialect of Adur Niesu, it lacks voiceless nasals /m̥/ and /n̥/. There are 10 monophthongs and one diphthong in Adur Niesu. A feature of Adur Niesu vowels is high vowel fricativization, occurring with the two high central vowels /i/ and /ɨ/, and the two high back vowels /u/ and /u̠/. Adur Niesu's syllable structure is relatively simple. All are open syllables. The following segmental changes are reported: vowel lowering, vowel centralization, vowel assimilation, vowel fusion, consonant lenition, and aspiration of clanlects. It is common for Adur Niesu syllables to be reduced in continuous speech. There are three main types of syllable reduction: complete reduction including the segment and tone, partial reduction with a floating tone left, and partial reduction with the initial consonant left. There are three contrastive tones in Adur Niesu, namely, high-level tone 55, mid-level tone 33, and low-falling tone 21. There is also a sandhi tone 44. There are two types of tonal alternation: tone sandhi and tone change. Tone sandhi occurs at both the word and phrasal levels, and is conditioned by the phonetic environment; tone change occurs due to the morphosyntactic environment. Moreover, some seeming tonal alternation is the result of the floating tone after syllable reduction.

**Funding:** This research received no external funding.

**Institutional Review Board Statement:** Not applicable.

**Informed Consent Statement:** Not applicable.

**Data Availability Statement:** Not applicable.

**Conflicts of Interest:** The authors declare no conflict of interest.

## Notes

[1]    Abbreviations: 1: first person, 2: second person, 3: third person, ATT: attitudinal maker, CLF: classifier, CLNK: clause linker, COM:comitative, CONT: continuous, CSM: change of state marker, DEF: definite, DOM: differential object marker, DSC: discourse clitic, EXCL: exclusive, EXST: existential verb, IMPF: imperfective, INTJ: interjection, LOC: locative, LOG: logophor, NEG: negation, NF: non-final marker, NMLZ: nominalizer, P: patient, PFV: perfective, PL: plural, POSS: possessive, QUOT: quotative, REDPL: reduplication, REP: repetitive, SEQ: sequential marker, SG: singular, SP: second part

[2]    It can refer to a grown-up animal, as long as it is some animal's offspring.

[3]    Adur Niesu resultative construction expresses the result happening to the affected entity, structured as affectee + instigator + complement clause, such as below. It does not follow the canonical SOV word order, but is construed in a topic–comment

| | | | | |
|---|---|---|---|---|
| lɛ$^{21}$bu$^{33}$ | tsʰi$^{33}$ | si$^{55}$ | dzɯ$^{33}$ | o$^{44}$. |
| cow:male | 3SG | kill | eat | PFV |

articulation.    'He killed the ox and ate (it).'

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
