# Peer review of "Phonology of Adur Niesu in Liangshan, Sichuan"

_languages, doi:10.3390/languages8030164_

Round 1

Reviewer 1 Report

The Nuosu languages, also known as Loloish or Yi in China, are a significant Tibeto-Burman subgroup of the Sino-Tibetan language family, distributed across various regions in China, Northern Vietnam, Northern Thailand, Northern Laos, East and Northern Burma, and Northeast India. The majority of Nuosu speakers inhabit southwestern China, particularly in the provinces of Sichuan, Yunnan, Guizhou, and Guangxi Zhuang Autonomous Region. These languages comprise various dialects, with some scattered across the aforementioned provinces and region. Notably, the northern dialects or vernaculars include Shynra (圣乍), Yynuo(义诺), Qumusu(Han-

Chinese: Tianba田坝), Suondi(索地), and Adur(阿都), among which Shynra is considered the standard language of the Yi people.

Within the northern pentad distribution of the Nuosu language, Shynra, Yynuo, and Qumusu are typically grouped together as Nuosu, while Suondi and Adur are classified as Niesu. Nuosu and Niesu, in turn, are grouped together as Nuosu proper. Shynra is the most well-researched dialect among the five, as it is the standard language. Suondi and Yynuo have also received considerable attention in research. However, Adur Niesu, the least researched of the five dialects in the northern Nuosu group, has recently become the focus of several studies, including this one, which aims to contribute to the scholarship on Nuosu languages by reporting on the phonology of Adur Niesu.

This article describes the segmental and suprasegmental phonology of Adur Niesu. It presents an inventory of consonants and vowels, phonotactics, syllable structure, and tone system of the language, as well as a thorough examination of phonological processes at both the segmental and suprasegmental levels. Comparisons are made with other Nuosu varieties, especially Suondi Niesu. The article is structured effectively, with explicit articulation of basic facts and well-grounded opinions and arguments.

As for questions and suggestions, the following are some for discussion:

To illustrate the value of this study, i.e., that this study contributes to the understanding of Adur Niesu, which is less adequately researched in the literature, the author has listed several works that focus mainly on Suondi Niesu, such as Mahai (2015, 2019) and Mise (2020), but another study on Adur Niesu (Sun, 2020) is not referenced. Sun's MA thesis from Southwest Minzu University in China, from which school in the same year Mise (2020) was graduated, reports on the construction of an Adur phonetics corpus and should not be overlooked as a work that focuses specifically on Adur Niesu.

As the author collected first-hand fieldwork data, a more detailed introduction to the background of the consultants and the time when the fieldwork was carried out would provide the reader with a better evaluation of the data and a more robust basis for comparison with other Nuosu varieties. Additionally, it is recommended to include Chinese characters for the two place names where Adur Niesu data are illustrated: “Tuojue(拖觉镇)” (Ln. 61) and “Jiaojihe(交际河)” (Ln. 128).

Given the similarities between Adur Niesu and Suondi Niesu, a description of the similarities and differences between these two varieties should be maintained for every part of the discussion of the phonology. For example, when describing the consonants and vowels inventory, it is desirable for readers to see how Adur is related to Suondi, so that they can understand the extent to which these two varieties share the segmental inventory and judge the similarities and differences between these two varieties, which bear the same name “Niesu” in opposition to “Nuosu”.

The author intends to illustrate lexical differences between Adur Niesu and Shynra Nuosu in Table 1 (Ln. 110). By examining the data in this table, a question arises about the nature of these differences. It seems that these words are related in the sense that they come from the same source of Proto-Loloish and are different in phonetic realization of the same root. Base on this understanding, their differences may be better labelled as “phonological” rather than “lexical”.

The title of Figure 6 (Ln .480) could be revised to “Figure 6. Adur Niesu tone exemplified by syllable [ma].”

There are some other minor typo or misprints. I am including them here for reference.

1. “昭觉县” (Ln. 41) is typesetted differently from other Chinese place names in the paper.

2. The two figures on P. 3 needs revising.

3. Ln. 198. The list numbering “1” should be “3”.

4. Ln. 267, “vowles” should be “vowels”

5. Ln. 338 “Error! Reference source not found”.

6. Ln. 472  “grammaticalor” should be “grammatical or”

7. Ln .493  “(ERROR! REFERENCE SOURCE NOT FOUND.43)”

8. Ln. 503  lexical sandhi rules in Adur Niesu are mostly occur in specific semantic

9. Ln. 518: “→ xɯ44dzɯ33” should be “→ ŋɯ44dzɯ33”

10. Ln. 552:  Could it be better to insert “gender” after the word “masculine” in “the masculine of the animate beings”?

11. Ln. 819:  lexical sandhi rules in Adur Niesu are mostly occur in specific

Reviewer 2 Report

 The author provides a detailed description and analysis of the sound system of the Adur Niesu language. The comparison of several generalization schemes for vowel phonemes, the analysis of various assimilation and dissimilation phenomena, and tonal alternations are excellent in the study of Yi languages. This article makes a tangible contribution to the documentation and study of Lolo-Burmese languages.

However, there are still some problems with this article, which are listed below.

1. What is the phonetic nature of the prenasalized plosives and affricates?, how do they need to be analyzed phonologically? Phonetically, is there a close relationship between the nasal part and the plosive/affricate part? Is there devoicing of the nasal part or of the plosive/affricate part? Phonologically, should the nasal part and the plosive/affricate part be analysed as a consonant cluster, or as a whole as a single consonant? This needs to be discussed in terms of phonotactics and economy.

2. Vowel assimilation should only include those with phonetic motivation. Those without phonological motives, or even those in which the occurrence of assimilation would distinguish the meaning, are more likely to be preservation of earlier phonetic features in larger speech segments, rather than synchronic assimilation phenomena. On this point, an attempt can be made to discuss the nature of this phonological alternation by comparing cognates in other varieties of Yi.

3. in Section 3.5, consonant “fortition” is, probably, also a preservation of earlier phonetic features in larger speech segments. Lenition does not require conditions, but fortition occurs only in certain syllable combinations. In addition, Malimasa (a Naish language) has /gɤ33/ 'afterwards'(https://zhongguoyuyan.cn/point/60448), which may be a cognate of ɣa44ɖʐɨ33 “afterwards”.

4. What is the phonetic nature of /ɯ/? Is it consistent across when combined with different consonant initials? Some previous studies have transcribed this vowel as [ʅ], which is not a standard IPA symbol, but suggests that the sound may have some specific properties. Also, it is relatively rare for there to be /i/-/ɿ(z̩)/-/ɯ/ oppositions in a language, and examples need to be given. If there are alternative possibilities for phonemic generalization concerning /ɯ/, these should also be discussed.

5. None of the various tone sandhi rules are across-the-board. From a phonological point of view, it may be necessary to consider the possibility of proposing more underlying tonal categories, or morphotonological rules, in the system (cf. Michaud’s (2017) book on Yongning Na). The tonal system may be too complex to be fully analyzed in one article, but the possibilities of alternative analysis should at least be mentioned, leaving these questions open for further research.
